



# Secular and orbital-scale variability of equatorial Indian Ocean summer monsoon winds during the late Miocene

Clara T. Bolton[1], Emmeline Gray[1,2], Wolfgang Kuhnt[3], Ann Holbourn[3], Julia Lübbers[3], Katharine Grant[4], Kazuyo Tachikawa[1], Gianluca Marino[4,5], Eelco J. Rohling[4,6], Anta-Clarisse Sarr[1], Nils Andersen[7]

1. Aix Marseille Univ, CNRS, IRD, INRAE, Coll France, CEREGE, Aix-en-Provence, France
2. Now at: School of Environment, Earth and Ecosystem Sciences, The Open University, Milton Keynes, UK
3. Institute of Geosciences, University of Kiel, D-24118 Kiel, Germany
4. Research School of Earth Sciences, Australian National University, Canberra ACT 2601, Australia
5. Centro de Investigación Mariña, Universidade de Vigo, GEOMA, Palaeoclimatology Lab, Vigo, 36310, Spain
6. Ocean and Earth Science, University of Southampton, Southampton SO14 3ZH, UK
7. Leibniz Laboratory for Radiometric Dating and Stable Isotope Research, University of Kiel, D-24118 Kiel, Germany

*Correspondence to*: Clara T. bolton (bolton@cerege.fr)

## Abstract

In the modern northern Indian Ocean, biological productivity is intimately linked to near-surface oceanographic dynamics forced by the South Asian, or Indian, monsoon. In the late Pleistocene, this strong seasonal signal is transferred to the sedimentary record as strong variance in the precession band (19-23 kyr) because precession dominates low-latitude insolation variations and drives seasonal contrast in oceanographic conditions. In addition, internal climate system feedbacks (e.g. ice-sheet albedo, carbon cycle, topography) play a key role in monsoon variability. Little is known about orbital-scale variability of the monsoon in the pre-Pleistocene, when atmospheric $CO_2$ levels and global temperatures were higher. In addition, many questions remain open regarding the timing of the initiation and intensification of the South Asian monsoon during the Miocene, an interval of significant global climate change that culminated in bipolar glaciation. Here, we present new high-resolution (< 1 kyr) records of export productivity and sediment accumulation from International Ocean Discovery Program Site U1443 in the southernmost Bay of Bengal spanning the late Miocene and earliest Pliocene (9 to 5 million years ago). Underpinned by a new orbitally-tuned benthic isotope stratigraphy, we use X-Ray Fluorescence-derived biogenic barium variations to discern productivity trends and rhythms. Our data show strong eccentricity-modulated precession-band productivity variations throughout the late Miocene, interpreted to reflect insolation forcing of summer monsoon wind strength in the equatorial Indian Ocean. On long timescales, our data support the interpretation that South Asian monsoon winds were already established by 9 Ma, with no apparent intensification over the late Miocene.



## 1 Introduction

The Asian monsoon is a major hydrological phenomenon driven by the pressure gradients created
by asymmetric heating between the equatorial Indian and western Pacific Oceans and the Indo-
Asian landmass, creating strong seasonally-reversing winds and ocean currents and heavy boreal
summer precipitation over the Bay of Bengal and Indian subcontinent (Webster, 1987a, b; Schott
and Mccreary Jr, 2001; Gadgil, 2003; Schott et al., 2009). Surface winds over the northern Indian
Ocean (Arabian Sea and Bay of Bengal) and South China Sea are strong indicators of the strength
of the South Asian and East Asian monsoon subsystems, respectively, and precipitation amount and
seasonality can also be diagnostic of monsoon strength (e.g., Webster and Yang, 1992; Goswami et
al., 1999). Thus, past monsoon dynamics can be reconstructed using wind, runoff, and precipitation
indicators recorded in marine sediments from these core convective regions. The Asian monsoon is
known to have varied substantially over short (interannual to suborbital) and long (orbital to
geological) timescales in response to forcing factors both external and internal to Earth's climate
system (e.g., Wang et al., 2005; Clemens and Prell, 2003; Farnsworth et al., 2019; Kathayat et al.,
2016).

There is great uncertainty surrounding the timing of Asian monsoon initiation and intensification,
and the degree of coupling between regional monsoon subsystems. Discrepancies in part stems
from the fact that records come from the South Asian or East Asian monsoon subsystems, which are
sensitive to different aspects of regional topography (Molnar et al., 2010; Clift et al., 2008; Clift and
Webb, 2019; Boos and Kuang, 2010; Acosta and Huber, 2020; Sarr et al., in review). Further,
differences in monsoon expression occur even within the core convective region of the South Asian
monsoon (e.g., dominance of summer monsoon winds in the southern Bay of Bengal *versus*
monsoonal rainfall/runoff in the northern and eastern parts). Meanwhile, proxies generally record
singular aspects of monsoonal climate that are not necessarily coupled on all timescales (e.g., winter
*versus* summer monsoon, wind intensity *versus* precipitation amount or seasonality). Evidence for
strong monsoonal climates (i.e., with strong seasonality of precipitation) exists across Asia during
the Paleogene (Spicer et al., 2017; Licht et al., 2014), yet many terrestrial records from southeast
Asia suggest an onset of the monsoon near the Oligocene-Miocene boundary (~24-22 Ma) (Guo et
al., 2002; Sun and Wang, 2005; Wang et al., 2005). Marine records of drift sedimentation near the
Maldives archipelago as well as upwelling and oxygenation indicators from the Arabian Sea, both
influenced by wind and surface ocean circulation, suggest an onset of strong seasonally-reversing
South Asian (monsoon) winds and Arabian Sea upwelling during the late middle Miocene (~13-10



Ma) (Zhuang et al., 2017; Gupta et al., 2015; Betzler et al., 2016; Betzler et al., 2018; Bialik et al., 2020; Nigrini, 1991).

In contrast, a time interval in which records from different regions and proxies converge somewhat
is the late Miocene. Magnetic records from the Chinese Loess Plateau are interpreted to show a
long-term intensification of the East Asian summer monsoon (EASM) from ~8.2-2.6 Ma (Ao et al., 2016). A late Miocene strengthening of Asian winter monsoons is inferred from South China Sea (Holbourn et al., 2018; Jia et al., 2003) and Andaman Sea (Lee et al., 2020) records. In the Arabian Sea, an intensification of upwelling and productivity at ~8 Ma is interpreted to reflect a
strengthening of the South Asian summer monsoon (SASM) (Kroon et al., 1991; Singh and Gupta, 2014; Gupta et al., 2004), although other studies find evidence contrary to this (Tripathi et al., 2017; Huang et al., 2007). Proposed monsoon intensifications during the late Miocene roughly coincide with strong global cooling (Herbert et al., 2016), and a number of studies have implicated cooling and the ramp-up of Antarctic glaciation in monsoon strengthening (Ao et al., 2016; Holbourn et al.,
2018; Gupta et al., 2004). Until now, a lack of continuous, well-preserved marine sequences from the South Asian monsoon region has stalled our understanding of its complex evolution during the Miocene.

The SASM is thought to have varied strongly on orbital timescales because monsoon strength
responds, both directly and via internal feedback mechanisms, to insolation forcing. Model simulations predict a stronger South Asian monsoon during summer insolation maxima at both precession minima and (to a lesser degree) obliquity maxima (Bosmans et al., 2018; Prell and Kutzbach, 1992, 1987; Kutzbach, 1981; Jalihal et al., 2019; Tabor et al., 2018). Precession is the dominant control on insolation and its seasonal distribution near the equator, and proxy-based
Pleistocene SASM records show strong precession-band (19-23 kyr) variability (Kathayat et al., 2016; Prell and Kutzbach, 1987; Clemens et al., 1991; Clemens et al., 2021; Zhisheng et al., 2011; Bolton et al., 2013; Caley et al., 2011; Gebregiorgis et al., 2018; Wang et al., 2005; Clemens and Prell, 1990; Rostek et al., 1997). The influence of global boundary conditions related to ice volume and greenhouse gas concentrations on SASM winds and precipitation/runoff is also demonstrated
by strong obliquity- and eccentricity-band variance in Plio-Pleistocene records (e.g., Clemens and Prell, 2003; Gebregiorgis et al., 2018; Clemens et al., 2021; An et al., 2011). Our current knowledge of orbital-resolution past productivity fluctuations in the South Asian monsoon region and their relationship with local (insolation) or remote (global ice volume, greenhouse gases) forcing mechanisms comes almost entirely from Pleistocene Arabian Sea sedimentary records (Clemens



and Prell, 2003; Caley et al., 2011; Singh et al., 2011; Shimmield and Mowbray, 1991; Rogalla and Andruleit, 2005; Clemens et al., 1991; Ziegler et al., 2010). These records suggest that summer monsoon proxies significantly lag northern hemisphere summer insolation maxima in the precession and obliquity bands due to climate feedbacks. Orbital control on past SASM strength in the pre-Pleistocene, when boundary conditions were different, has so far only been investigated in

the Andaman Sea over the latest Miocene-early Pliocene, where seawater oxygen isotope data suggest high-amplitude precession and obliquity forcing of monsoon rainfall with significant phase lags (Jöhnck et al., 2020).

In this paper, we investigate sediment accumulation and export productivity dynamics at millennial

resolution in late Miocene sediments from southern Bay of Bengal (BOB) International Ocean Discovery Program (IODP) Site U1443 (5°N, 90°E, Fig. 1). The late Miocene (11.6-5.3 Ma) is an interval of major global climate change, with long-term cooling between ~7.5 and 5.5 Ma (Herbert et al., 2016) culminating in major high-latitude cooling events (Holbourn et al., 2018), and important carbon cycle shifts recorded in the marine and terrestrial realms potentially linked to

atmospheric $CO_2$ decline (Tauxe and Feakins, 2020; Steinthorsdottir et al., 2020). The region of Site U1443 is strongly influenced by seasonally reversing monsoon winds today, and primary productivity is tightly coupled to the annual monsoon cycle (Fig. 1a-b, Fig. 2). During the SASM, strong moisture-laden winds blow inland, driving surface circulation changes and increased mixed layer depth (Fig. 1a-b) (Webster, 1987a, b; Schott and Mccreary Jr, 2001; Gadgil, 2003; Schott et

al., 2009). Strong Ekman pumping mixes nutrients into the surface layer during the South Asian summer monsoon and, to a lesser extent, the winter monsoon, stimulating biological productivity (Fig. 2) (Lévy et al., 2007; Mccreary et al., 2009; Koné et al., 2009; Behrenfeld et al., 2005; Longhurst, 1995). This strong seasonal signal is transferred to the sedimentary record in the form of strong variance at orbital periods because insolation variations drive seasonal contrast. In this paper,

we generate a new orbitally tuned age model based on benthic foraminiferal stable isotopes spanning ~9 to 5 Ma. Core-scanning X-Ray Fluorescence (XRF) data are then used to reconstruct bulk, carbonate, and biogenic barium content and mass accumulation rates (MARs), shedding light on secular and orbital-scale export productivity and sedimentation changes over the late Miocene.

**2 Background**

In the modern southern BOB in waters overlying Site U1443, both primary and export productivity are strongly controlled by seasonally reversing winds associated with the South Asian monsoon.



Figure 2 shows the annual cycle of wind stress, mixed layer depth (MLD), net primary productivity
(NPP), and biogenic particle export based on recent oceanographic, satellite, and sediment trap data
(see methods for details). In boreal summer (JJA) strong southwest winds mix the upper water
column, deepening the MLD to ~60m and entraining nutrients into the photic zone, leading to
enhanced primary productivity and biogenic particle export (with a lag of ~3-4 weeks) (Fig. 1a-b,
Fig. 2). During boreal winter, northeast winds deepen the MLD to a lesser extent, resulting in a
second smaller peak in productivity during the winter monsoon (DJF) (Fig. 2). During the inter-
monsoon seasons, lowest wind stress is recorded leading to a shallow MLD, higher Sea Surface
Temperatures (SSTs), and a more stratified upper water column, resulting in increased oligotrophy
and reduced biological productivity. The biannual productivity maxima observed in the surface
ocean above Site U1443 is characteristic of monsoon-dominated tropical regions (Longhurst, 1995;
Lévy et al., 2007). Sediment trap data suggest that primary productivity is the dominant control on
organic carbon export at a location west of the northern end of the Ninetyeast Ridge (SBBT site;
5°N, 87°E, Fig. 1c). However, lithogenic mineral ballasting at this location is not negligible
(average ~13% and 15% lithogenic particles in shallow and deep traps respectively) (Rixen et al.,
2019; Unger et al., 2003) and could in part explain the bias towards the late summer peak seen in
biogenic fluxes compared to NPP, as maximum concentrations of lithogenic particles at SBBT
occur during the summer monsoon. While wind forcing is identified as the dominant factor
controlling biogenic particle fluxes at the SBBT site, advection of nutrient- and chlorophyll-rich
waters originating from the eastern Arabian Sea via the Southwest Monsoon Current (SMC) may
further contribute to the summer productivity peak in this region (Unger et al., 2003). During the
summer monsoon, the relatively salty and nutrient-rich SMC flows eastwards south of Sri Lanka
then turns northwards into the BOB (Fig. 1c) (Schott et al., 2009; Jensen, 2003). The SMC and
associated eddies have been shown to increase chlorophyll concentrations and average
phytoplankton size along their paths as far east as 88-90°E, with the current's influence generally
restricted to north of 6°N at this longitude (Jyothibabu et al., 2015; Vinayachandran et al., 2004;
Webber et al., 2018). While river runoff and resultant salinity stratification during the summer
monsoon supress primary productivity further north in the BOB (Prasanna Kumar et al., 2002),
seasonal surface salinity variations are very small (<0.2 psu) at 5°N (Zweng et al., 2013).
Accordingly, monsoon impacts on nutrients and productivity in our study area are limited to those
driven by surface currents and wind mixing, and biogenic export fluxes during the SASM are
similar (Particulate organic carbon, POC) or higher ($CaCO_3$ and biogenic $SiO_2$) than at sites further
north in the BOB (Unger et al., 2003). Thus, modern data give us confidence that export
productivity at our site is likely a reflection of South Asian (primarily summer) monsoon wind



strength, via its control on MLD and nutrient entrainment into the mixed layer as well as on surface ocean currents.

## 3 Materials & Methods

### 3.1 Site and sampling

Samples used in this study are from Site U1443 (Latitude 5°23 N, Longitude 90°21 E, water depth 2935 m), drilled during IODP Expedition 353 on the crest of Ninetyeast Ridge (NER) (Clemens et al., 2016) (Fig. 1c). Site U1443 is located ~100 m southeast of Ocean Drilling Program (ODP) Site 758 and is a re-drill of this legacy site (Shipboard Scientific Party, 1989). At Site U1443, the use of Advanced Piston Coring (APC) and half-length APC drilling techniques down to >200 m CSF (core depth below sea floor) in three holes allowed recovery of a compete, spliced Neogene sedimentary section spanning 0-195 m CCSF (core composite depth below sea floor). Late Miocene records cover the interval 70.06 m CCSF (U1443B 7H 5W 75-76 cm) to 122.76 m CCSF (U1443C 15H 4W 148-149 cm), following the revised shipboard splice (CCSF-D), spanning the interval ~5 Ma to 9.5 Ma based on initial bio-magneto-stratigraphy. Samples come from lithologic Units Ib and IIa, and sediments mainly consist of light grey to pale yellow nannofossil ooze with clay and foraminifers, and occasional volcanic ash (Clemens et al., 2016). To minimise aliasing due to low estimated sedimentation rates (~0.5-2 cm/kyr), 1 cm thick half rounds (~18 cc) were sampled for micropaleontology. Cores were sampled at a depth resolution of 4 cm in the upper part of the study interval (70.06-114.18 m) and 2 cm in the lower part (114.18-122.76 m) where sedimentation rate estimates were lower. U-channels for XRF scanning were sampled from archive halves of sediment cores at Kochi Core Centre (Japan) during the post-cruise sampling party for 39 sections included in the splice between U1443C 9H 2A (69.95 m CCSF) and U1443C 13H 5A (113.58 m CCSF).

Modern oceanographic conditions over the seasonal cycle above Site U1443 were assessed using recent datasets (Figs. 1a-b, 2). Monthly data for wind (Wind Stress, Metop-A ASCAT, 0.25°, Global, Near Real Time, 2009-present) (Fig. 2a), MLD (1969-2010) (Keerthi et al., 2013) (Fig. 2b) and depth-integrated NPP estimated from satellite-derived surface chlorophyll concentrations (Primary Productivity, Aqua MODIS, NPP, Global, 2003-present, EXPERIMENTAL (Monthly Composite) calculated using method of Behrenfeld and Falkowski (1997); (Erd, 2020)) (Fig. 2c) were extracted for a box between ~4.5-5.5°N latitude and 89-91°E longitude (depending on grid resolution) and binned by month. Scatter thus reflects a combination of spatial variability within our small box and interannual variability; monthly means over each time series are also shown. POC,



biogenic $SiO_2$, and $CaCO_3$ fluxes (Fig. 2d, e) are from SBBT sediment trap samples (~5°N, 87°E,

Fig. 1c) (Unger et al., 2003). In Fig. 2 we show data from the deep traps (~3000 m, ~21 day

sampling intervals) (Rixen et al., 2019), but seasonal patterns of biogenic particle flux are very

similar in the shallow (~1000 m) SBBT traps (Unger and Jennerjahn, 2009; Vidya et al., 2013).

Data points show fluxes recorded in individual years (1987-1997, plotted against mid-time for the

trap deployment), with monthly averages also shown. Monthly wind and NPP data were

downloaded from the ERDDAP server (https://coastwatch.pfeg.noaa.gov/erddap/index.html) and

Indian Ocean MLD data (Keerthi et al., 2013) from

http://www.ifremer.fr/cerweb/deboyer/mld/Surface_Mixed_Layer_Depth.php.

**3.2 Late Miocene benthic foraminiferal stable isotope data**

Bulk sediment samples were oven-dried at 50°C, weighed, and washed over a 63 µm sieve in tap

water at CEREGE. The >63 µm fraction was oven-dried at 50°C on a filter paper and weighed to

determine percentage coarse fraction. The <63 µm fraction was centrifuged and dried at 50°C.

Depth resolution for the benthic isotope record is 8 cm (70.06 m to 112.87 m CCSF), 4 cm (112.86

m and 114.18 m CCSF) or 2cm (114.18-122.76 m CCSF), except near splice tie-points where

sampling included overlap between cores increasing resolution. Six to twelve specimens of the

epibenthic foraminiferal species *Cibicidoides wuellerstorfi* were picked from the >212µm fraction,

with 6-8 well-preserved specimens selected for analysis. Tests were broken into fragments, cleaned

in ethanol in an ultrasonic bath, and oven dried at 40°C. Stable carbon and oxygen isotopes were

measured on a Thermo Scientific MAT 253 dual-inlet isotope ratio mass spectrometer (DI-IRMS)

coupled to Kiel IV carbonate preparation device at the Leibniz Laboratory, University of Kiel.

Based on long-term analysis of international and internal carbonate standards, precision (1σ) is

better than ±0.08‰ for $\delta^{18}O$ and 0.05‰ for $\delta^{13}C$. Results were calibrated using the National

Institute of Standard and Technology (NIST) carbonate isotope standard NBS (National Bureau of

Standard) 19, and are reported on the Vienna PeeDee Belemnite (VPDB) scale. *C. wuellerstorfi*

isotope data below 112.86 m CCSF were originally published in Lübbers et al. (2019), and a low-

resolution version of the long-term $\delta^{13}C$ record is included in Bretschneider et al. (submitted).

   **3.3. Age model**

Seven calcareous nannofossil bio-events dated between 5.04 Ma and 9.53 Ma were identified at

high-resolution in Site U1443 splice samples (Table S1) to increase the resolution of shipboard

biostratigraphy (Robinson et al., 2016). To check for orbital periodicities prior to tuning, wavelet



analyses were performed on benthic isotope records in the depth domain and on the revised nannofossil-based age model (using a 4th order polynomial fit, Fig. S1) in R using the biwavelet

package (Gouhier et al., 2016; Grinsted et al., 2004) (Fig. S2a-d). All time-series were first interpolated to constant depth or age resolution, such that the maximum resolution present was preserved (2 cm and 2 kyr for benthic isotope records in the depth and age domain, respectively). Records were then filtered to remove signals with periods longer than one third of the length of the dataset using the "bandpass" function in the R package Astrochron (Meyers, 2014). Wavelets

indicate that obliquity-driven cycles are present throughout (~0.53 m, Fig S2 a-b; 41 kyr and 53 kyr, Fig. S2 c-d), confirming that the U1443 record is suitable for orbital tuning. Using revised nannofossil datums (Table S1, Fig. S1) and shipboard magnetostratigraphy (Clemens et al., 2016) as preliminary age-depth tie-points (Fig. 3), an astronomical age model was constructed by tuning our monospecific benthic $\delta^{18}O$ record to an eccentricity plus tilt (ET) target curve (Laskar et al.,

2004) (Fig. 4). We did not include precession in our tuning target so as not to introduce assumptions related to which hemisphere was controlling climate at our site, and because the temporal resolution of our benthic record does not permit accurate resolution of precession cycles in some intervals. We used a minimal tuning approach, tying ET maxima to benthic $\delta^{18}O$ minima, with at most one tie-point per ~100 kyr and often one every 200-300 kyr (Fig. 4, Fig. S1), so as not to artificially

introduce frequency modulations (Zeeden et al., 2015).

### 3.4 XRF Scanning and Calibration

U-channels were scanned at 1 cm intervals at The Australian National University (ANU) on a third generation Avaatech XRF core scanner. All cores were scanned sequentially and standards

measured daily were consistent across all runs. Core sections were covered with 4 micron-thick Ultralene film and measured at 10 kv with a 500 µA current and no filter, then at 30 kv with a 200 µA current and Pd thin filter, and finally at 50 kv with a 50 µA current and Cu filter. A 30 sec count time was used for all runs. Late Miocene XRF data generated at ANU (72.75- 113.56m CCSF) were spliced with data from Lübbers et al. (2019) (112.80m to 122.76m CCSF), also scanned on an

Avaatech XRF core scanner but with different machine settings. To splice the records, we rescaled the raw Lübbers et al. (2019) elemental count data so that absolute values and variance matched our data, based on an overlapping interval between 112.80 and 113.56m CCSF.

Quantitative chemical compositions of a subset of discrete bulk sediment samples were determined

at CEREGE after total digestion by Inductively Coupled Plasma Mass Spectrometer (ICP-MS



Agilent 7500 ce). Twenty samples from the scanned late Miocene interval, selected to cover the range of values in the raw XRF count data for elements of interest, were analysed and concentrations of Al, K, Ca, Ti, Mn, Fe, Rb, Sr, and Ba were determined. Prior to analysis, samples were dried and homogenised in a pestle and mortar. About 30 mg of sediment was completely

dissolved by acid digestion using a 2:1 mixture of ultrapure acids (15 M $HNO_3$ and 22 M HF with $HClO_4$) on a hot plate. Blank contribution was estimated to be negligible. The accuracy of measurements was evaluated using analysis of geostandards MAG-1(marine mud) and BE-N (basalt). The typical analytical uncertainty was better than 5%. XRF-derived element counts were converted into element concentrations by direct linear calibration. This allowed us to reduce

uncertainties related to the variable matrix effect and physical properties such as moisture content that typically change downcore. Additionally, we used calibrated XRF data to calculate biogenic barium concentrations and to estimate % $CaCO_3$ and "carbonate-free basis" (cfb) elemental concentrations, permitting evaluation of the extent to which dilution by the dominant sediment constituent (here $CaCO_3$) is driving trends and variability of more minor constituents in our records.

To represent the relative contributions of $CaCO_3$ versus terrigenous sediment components, we use the log count ratio of $Ca/(\sum(\text{Al, K, Ti, Fe, Rb}))$, termed log(Ca/Terr).

### 3.5 Ba-based export productivity proxies

The accumulation of biogenic barium in sediments is a reliable proxy for export production in

certain environments (Paytan and Griffith, 2007). Micron-sized barite ($BaSO_4$) crystals are the main carriers of particulate barium in the ocean, with a maximum in concentration occurring just below the euphotic zone (Bishop, 1988; Dehairs et al., 1980). Although the exact mechanisms governing the precipitation of barite in the water column are only now coming to light (Martínez-Ruiz et al., 2019), its formation is thought to be associated with decaying organic matter. Depth profiles of

dissolved Ba suggest that passive adsorption of barite onto mainly biogenic particles as they sink through the water column, combined with vertical mixing of dissolved Ba from the deep ocean and riverine input, can best explain Ba's nutrient-like water column distribution (Dehairs et al., 1980; Cao et al., 2016). Goldberg and Arrhenius (1958) first hypothesised that an increase in Ba accumulation rate in sediments underlying the equatorial Pacific divergence zone was directly

linked to overlying high productivity, followed by similar observations in equatorial Indian Ocean sediments (Schmitz, 1987). Subsequently, evidence for strong correlations between fluxes of Ba and organic carbon in Atlantic and Pacific sediment traps led to algorithms relating new productivity to particulate Ba flux (Dymond et al., 1992; Francois et al., 1995). A further study





focusing on the accumulation of barite ($Ba_{barite}$) extracted from core-top and late Pleistocene

sediments refined its use as a proxy for export productivity (Paytan et al., 1996). Although

significant Ba regeneration occurs in the uppermost few millimetres of sediment (Paytan and

Kastner, 1996), barite dissolution is thought to cease after burial due to supersaturation in interstitial

waters (Gingele and Dahmke, 1994; Dymond et al., 1992) and barite is not subject to burial

diagenesis in oxic sediments (Paytan et al., 1993). Ocean sedimentary Ba has both a biogenic

($Ba_{bio}$) and a terrigenous ($Ba_{detrital}$) component, so estimates of past export productivity using barium

must distinguish between these sources. This can either be done by chemical leaching of bulk

sediment (assuming that all barite is $Ba_{bio}$) e.g. (Paytan et al., 1996), or by determination of total

barium ($Ba_{total}$) and subtraction of $Ba_{detrital}$ using Al content and the terrigenous Ba/Al ratio,

resulting in a record of $Ba_{xs}$ ($Ba_{total} - Ba_{detrital}$), see equation 1 (Dymond et al., 1992).


$$[Ba_{xs}]_{ppm} = [Ba_{total}]_{ppm} - (Ba/Al_{terrigenous})*[Al_{total}]_{ppm} \quad (1)$$

Direct comparisons of measurements of $Ba_{barite}$ and $Ba_{xs}$ suggest that non-barite phases of barium

may be included in the calculation of $Ba_{xs}$; nevertheless $Ba_{xs}$ is most representative of $Ba_{barite}$ and

therefore export productivity in oxic carbonate-rich sediments with low terrigenous, biogenic silica,

and organic carbon contents (Eagle et al., 2003; Averyt and Paytan, 2004; Gonneea and Paytan,

2006). The use of bulk Ba/Ti, Ba/Al, and Ba/Fe ratios is another approach to evaluate relative

changes in export productivity (i.e., normalisation to an element presumed to be predominantly of

terrigenous origin), but without precisely predefining the Ba/terrigenous ratio that could vary over

time, and also removing the effect of dilution by a dominant sedimentary component such as

$CaCO_3$ (Murray et al., 2000).

Here, we reconstruct changes in export productivity at Site U1443 over the late Miocene using

XRF-derived Ba data and compare elemental count ratios of log(Ba/Fe), log(Ba/Ti), and

log(Ba/Al), with $[Ba]_{cfb}$ and $[Ba]_{xs}$ calculated following equation (1), using a Ba/Al$_{terrigenous}$ value of

0.0075 g/g following Dymond et al. (1992). To verify consistency of trends, we also calculate

$[Ba]_{xs}$ using [Ti] to represent $Ba_{detrital}$, applying a Ba/Ti$_{terrigenous}$ ratio of 0.183 g/g (Mclennan, 2001),

and carbonate-free $[Ba]_{xs}$.



### 3.6 Mass Accumulation rates

MARs of bulk sediment, $CaCO_3$, $[Ba]_{xs}$, and summed terrigenous elements (Al, K, Ti, Fe, and Rb) were calculated by multiplying concentrations by linear sedimentation rates (in cm/kyr) derived from our new age model and dry bulk densities (in $g/cm^3$). Dry bulk density values were estimated from high-resolution shipboard Gamma Ray Attenuation bulk density scanning data, and the linear relationship between all shipboard U1443 wet bulk density and dry bulk density measurements ($n=164$, $r^2 > 0.99$) (Clemens et al., 2016). Units are $g/cm^2/kyr$ for bulk and $CaCO_3$ MAR, $\mu g/cm^2/kyr$ for $[Ba]_{xs}$ MAR and $mg/cm^2/kyr$ for terrigenous MAR.

### 3.7 Time series analysis

Spectral analyses of benthic isotope and XRF data against tuned age were performed on filtered records with a constant time step as described in Section 2.3 (0.5 kyr for XRF records and 2 kyr for isotope records). Multi-taper method (MTM) spectral analyses using a robust red-noise model were performed using Acycle (Li et al., 2019). Blackman-Tukey cross-spectral analyses were performed in Arand to assess phase and coherence (Howell et al., 2006). Wavelet analyses were performed in R using the biwavelet package (Gouhier et al., 2016; Grinsted et al., 2004). To illustrate precession-band variance and amplitude modulation, certain records (with identified significant precession variance) were filtered using a Tanner-Hilbert filter centred on 43 cycles/myr with bandwidth ±3 (designed to include both the ~24 kyr and ~22 kyr precession periods) in Acycle (Li et al., 2019).

## 4. Results

### 4.1 Age model and benthic foraminiferal isotope data

U1443 benthic $\delta^{18}O$ and $\delta^{13}C$ data between 70.06 m and 122.76 m CCSF are shown in the depth domain in Figure 3 alongside calcareous nannofossil datums (revised herein, Table S1) and magnetochron boundaries. Our tuned benthic $\delta^{18}O$ and $\delta^{13}C$ records, shown in Figure 4, span the interval 4.96 Ma to 8.99 Ma and our age model shows excellent agreement with revised biostratigraphic and shipboard magnetostratigraphic datums (Fig. S1). Sedimentation rates generally vary between 1 and 1.7 cm/kyr, with a minimum of ~0.5-0.7 cm/kyr in the oldest part of the record (8.6 to 9 Ma) and a maximum of ~1.9 cm/kyr at 7.8-8 Ma (Fig. 4d). We note that between 112.86 m and 121 m CCSF (8.7-8.1 Ma), our age model differs by up to 60 kyr from that of Lübbers et al. (2019), which is based on correlation of Site U1443 benthic $\delta^{18}O$ and $\delta^{13}C$ data to





the orbitally tuned ODP Site 1146 δ¹³C record (Holbourn et al., 2018) using a limited number of tie
points (Site locations in Fig. 1c). Wavelet analyses (Fig. 4 f, g) as well as spectral analyses (Fig.
S2e, f) of the tuned benthic records reveal significant orbital periodicities (>99% significance level)
of ~405 kyr and 41 kyr for δ¹⁸O and ~405 kyr, 125 kyr, 95 kyr, 53 kyr and 41 kyr for δ¹³C, and
filtered isotope records show good correspondence with filtered ET (Fig. 4h). Cross-spectral
analysis between δ¹⁸O and δ¹³C reveals >95% coherency in the 41 kyr and 405 kyr bands (Fig.
S3a). Our age model is supported by close agreement between Site U1443 benthic δ¹⁸O and δ¹³C
data and independent orbitally-tuned benthic isotope records from the South China Sea (ODP 1146)
(Holbourn et al., 2018; Holbourn et al., In Press), equatorial Pacific (IODP Sites U1338 and 1337)
(Drury et al., 2016; Drury et al., 2018; Drury et al., 2017), and equatorial Atlantic (ODP Sites 926
and 999) (Bickert et al., 2004; Shackleton and Hall, 1997; Drury et al., 2017; Zeeden et al., 2013)
(Figs S4, S5).

Mean temporal resolution of the Site U1443 benthic isotope record is 4.2 kyr. Between 9 Ma and
7.6 Ma, mean benthic δ¹⁸O values vary between 2.5 and 2.8 ‰, with an overall decreasing trend
culminating in minimum values averaging ~2.4 ‰ between 7.6 and 7 Ma (Fig. 4c). Between 7 and
6.5 Ma, mean δ¹⁸O values increase by ~0.25 ‰, and between 6.5 and 5 Ma, mean values vary
between 2.55 and 2.8 ‰. Between 6 and 5 Ma, a number of prominent benthic δ¹⁸O maxima are
identified in the U1443 δ¹⁸O record, namely TG2, TG12, TG14, TG20 and TG22 (following
nomenclature of Shackleton et al. (1995)) (Fig. 4c). Between 7.7 and 6.9 Ma, strong obliquity
modulation of the U1443 δ¹⁸O record is seen (Fig. 4f), as also noted at Sites U1337 (Drury et al.,
2017) and 1146 (Holbourn et al., 2018) (Fig. S4). Long-term trends are similar to those recorded at
Pacific sites, with benthic δ¹⁸O values at Indian Ocean Site U1443 ~0.1‰ heavier than at Pacific
Sites U1337/U1338 and ~0.2-0.3‰ heavier than at South China Sea Site 1146 (Fig. 5a).


Mean benthic δ¹³C values at Site U1443 vary between 0.7 and 1.1‰ from 9 to 7.6 Ma, then
decrease from ~1 to -0.2‰ between 7.6 and 6.7 Ma, reflecting the globally recognised Late
Miocene Carbon Isotope Shift (LMCIS) (Keigwin, 1979; Keigwin and Shackleton, 1980) (Fig. 4e).
From 6.7 to 5 Ma, mean values vary between -0.2 and 0.4 ‰. The timing of the LMCIS at Site
U1443 (~7.6 Ma to ~6.7 Ma) is synchronous with the event in independent orbitally-tuned high-
resolution records (Drury et al., 2018; Drury et al., 2017; Holbourn et al., 2018; Drury et al., 2016)
(Fig. 5b, Fig. S5), and its magnitude (~1‰ decrease in δ¹³C in smoothed record) is similar to that
recorded in Pacific Ocean sediments from Sites U1338, U1337 and 1146. The Site U1443 δ¹³C



record shows a consistent positive offset of 0.15-0.25 ‰ relative to South China Sea Site 1146 over
the 9-5 Ma interval (Fig. 5b).

**4.2 XRF data and calibration**

Scanning XRF results are shown in Figure 6. Linear calibration between element counts and
concentrations in discrete samples over the late Miocene interval (~8.15 to 5 Ma) showed
significant coefficients of determination, with $R^2$ values ranging from 0.68 (Al) to 0.87 (Fe) (Fig.
S6), excluding Sr and Ca (see explanation below). Raw and calibrated elemental data show
consistent trends and amplitude variability (Fig. 6). For Ti, Ba, Al, and Mn, the re-scaled counts/sec
values in the 113.37-122.76 m CCSF interval (Lübbers et al., 2019) fell outside of our calibration
range, thus data below 113.6m (~8.15 Ma) were not converted to concentrations (Fig. 6). In brief,
Al, Si, Ti, Fe, Rb, and K show similar trends, with a long-term small increase in concentrations
from 8.15 Ma to 5 Ma and spikes (particularly pronounced in Rb and K) corresponding in some
cases to described ash layers (Clemens et al., 2016). Ca and Ba show a minor long-term decrease
over the study interval, while Sr and Mn increase from ~8.15 Ma to 6 Ma, then stabilise or decrease
slightly. All elements show high-frequency variability throughout. For Ca and Sr, $R^2$ values were
lower (0.39 and 0.42 respectively, Fig S6) due to the consistently high Ca and Sr contents and small
variability in the selected calibration samples. To estimate percent $CaCO_3$, we therefore used a
Ca/Fe ratio calibration rather than a direct linear calibration. We first used the linear relationship
between Ca/Fe counts and Ca/Fe as determined by ICP-MS (Fig. S6, $R^2$=0.93). Then %$CaCO_3$ was
calculated assuming that all Ca was contained in $CaCO_3$ – a reasonable assumption at Site U1443
given the relatively low clay content in lithological subunits Ib and IIa (Clemens et al., 2016).
Confidence in our method is provided by very good agreement with independent %$CaCO_3$
estimates for the middle and early late Miocene interval of Site U1443 based on calibration of XRF-
derived counts of (Ca/$\sum$(Ca, Al, Si, K, Ti, Mn, Fe, S)) to discrete $CaCO_3$ measurements (Lübbers et
al., 2019), including an overlapping interval based on an alternate splice from 112.8 and 113.6 m
CCSF (Fig. 7b).

**4.3 $CaCO_3$ content, sediment accumulation patterns and Ba proxies**

Late Miocene estimated % $CaCO_3$ varies between ~60 and 90% with a small long-term decrease
over the 9-5 Ma interval (Fig. 7b). This long-term trend is also visible in the log(Ca/Terr) record
(Fig. 7c), implying a small increase in the contribution of terrigenous material relative to $CaCO_3$ in
Site U1443 sediments over time. Three % $CaCO_3$ and log(Ca/Terr) minima between 5 and 6 Ma



occur in identified ash layers. Log(Ba/Al), log(Ba/Fe) and log(Ba/Ti) show identical long-term and orbital-scale trends (Fig. S7), therefore we only discuss log(Ba/Fe) in the main text. Log(Ba/Fe) shows a long-term decrease between 9 and 5.3 Ma, and a smaller increase from 5.3 to 5 Ma (Fig.

7d). $[Ba]_{xs}$ shows identical variability whether calculated using Al or Ti (Fig. 7e), and generally shows similar patterns to log(Ba/Fe) where records overlap (8.15 to 5 Ma). Values of $[Ba]_{xs}$ generally vary between 400-800 ppm, however carbonate-free $[Ba]_{xs}$ concentrations are typically 1000 to 4000 ppm (Fig. S7). A peak in log(Ba/Fe) between 7.6 and 7.3 Ma is less pronounced in the $[Ba]_{xs}$ record, but is prominent in the carbonate-free $[Ba]_{xs}$ record (Fig. S7, grey shading),

suggesting that this peak is supressed in the $[Ba]_{xs}$ record as a result of dilution by carbonate. The trough between 7.6 and 7.9 Ma seen in log(Ba/Fe), $[Ba]_{xs}$ and to a lesser extent in % $CaCO_3$, log(Ca/Terr), and carbonate-free $[Ba]_{xs}$ appears not to be an artefact of dilution.

Bulk sediment MARs vary between 0.5 and 2.1 $g/cm^2/kyr$ with a step increase from ~0.5 to 1.5

$g/cm^2/kyr$ occurring at 8.66 Ma (Fig. 7g), concurrent with a major sedimentation rate increase (Fig. 4d). $CaCO_3$ MARs range from 0.4 to 2 $g/cm^2/kyr$ and co-vary with bulk sediment MARs, with the increasing difference between the two records reflecting a small long-term increase in non-$CaCO_3$ components (Fig. 7g). This small increase is reflected in terrigenous element MARs, which vary between ~20-80 $mg/cm^2/kyr$ (excluding volcanic ash layers) (Fig. 7i). We note that absolute values

of terrigenous MAR should be interpreted with caution, because this calculation does not include Si as this element was not quantified in discrete samples. Nevertheless, a significant correlation between Al and Si counts ($R^2 = 0.8$, Fig. 6) suggests that Si is primarily of terrigenous origin, therefore trends in log(Ca/Terr) and terrigenous MAR in Fig. 7 are likely robust despite the exclusion of Si. From 8.3-5 Ma, $[Ba]_{xs}$ MAR shows similar patterns to $CaCO_3$ MAR, with no clear

long-term trend and maximum values driven by higher sedimentation rates in the intervals 5-5.2 Ma, 6.1-6.3 Ma, 7.5-7.7 Ma, and 7.8-8 Ma (Fig. 7f).

Spectral analyses reveal significant orbital periods in all late Miocene XRF records (Fig. 8). The ~405 kyr period is >99% significant in [Al], [Ba], $[Ba]_{cfb}$, $[Ba]_{xs}$, and log(Ba/Fe), whereas the ~125

kyr period is significant (>95% or >99%) in [Ba], $[Ba]_{cfb}$, $[Ba]_{xs}$, log(Ba/Fe), and % $CaCO_3$ records. At higher frequencies, the spectral signatures of [Fe], [Al], [Ti], [Ba], are dominated by significant peaks at 24 kyr (>99%) and 41 kyr (>90%), with [Ba] additionally showing peaks at 22.5 kyr (>99%) and at 26 and 30 kyr (>95%). Log(Ba/Fe), $[Ba]_{cfb}$, and $[Ba]_{xs}$ show dominant (>99% significant) 22.5 kyr variability, with additional >95% significant peaks at 24 kyr and 30 kyr (for

$[Ba]_{xs}$ only). Log(Ca/Terr) shows significant peaks at 24 and 22.5 kyr (both >99%), and also at 68



kyr (>95%). Percent CaCO$_3$ contains significant (>95%) variability at 22.5 kyr, with additional peaks at 68 kyr and 37 kyr. In summary, all records show highly significant variability in the precession band (22-24 kyr), with variability at the 22.5 kyr period and the ~125 kyr period most strongly associated with the biogenic component of Ba and with CaCO$_3$. Wavelet analyses of

log(Ba/Fe) and [Ba]$_{xs}$ confirm significant precession-scale variability in these records throughout the 9-5 Ma interval (Fig. 9).

## 5. Discussion

### 5.1 Late Miocene sedimentation patterns in the southern Bay of Bengal

We first examine the drivers of changes in sediment MAR identified in our record, and their possible link to regional and global productivity trends. The three-fold increase in CaCO$_3$ MAR at 8.66 Ma at Site U1443, originally described in Lübbers et al. (2019), could result from improved preservation and/or increased carbonate export by pelagic calcifiers (coccolithophores and/or foraminifera). Based on CaCO$_3$ percentages, MARs, and benthic to planktic foraminiferal ratios,

Lübbers et al. (2019) identified the mid to late Miocene "carbonate crash" in Site U1443 sediments between ~12.2 Ma and 10 Ma, with a slow recovery from ~10 to 8.7 Ma, favouring an interpretation that the increase in CaCO$_3$ MAR at 8.66 Ma reflects improved preservation. A record of planktic foraminiferal fragmentation between 9 and 8 Ma generated in the present study, interpreted to reflect a decrease in carbonate dissolution (Le and Shackleton, 1992), supports this

interpretation (Fig. 7h). We see no change in log(Ba/Fe) concurrent with the CaCO$_3$ MAR increase at 8.66 Ma, which suggests that total export productivity at Site U1443 remained stable over this transition. However, our data suggest that an increase in coccolithophore production may have occurred. The contribution of foraminifera to total CaCO$_3$ over our study interval is low (see >63 µm MAR in Fig. 7g), leading us to infer that higher CaCO$_3$ MARs between 8.66 and 5 Ma are

primarily driven by coccoliths. A 3-fold increase in sediment accumulation rate (SAR) at ~8.6 Ma with no change in CaCO$_3$ content (%), implying a large increase in CaCO$_3$ MARs, is also seen at shallower (2247 m) Deep Sea Drilling Project (DSDP) Site 216 on the NER near the equator (Fig. 1c) (Bukry, 1974; Mcneill et al., 2017; Pimm, 1974). This suggests that production played a role in driving regional carbonate MAR increases as well as improved preservation at deeper sites. A

recent study decoupling coccolith and foraminiferal MARs in relatively shallow, globally-distributed sites (minimally affected by dissolution) records a late Miocene pulse in coccolith MARs beginning at ~8-9 Ma and persisting until ~3-4 Ma, interpreted to reflect high



coccolithophore productivity and calcification driven by weathering alkalinity inputs and regional nutrient changes (Si and Rosenthal, 2019).


Interestingly, the increase in bulk MARs at 8.66 Ma is driven by both $CaCO_3$ and to a lesser extent non-$CaCO_3$ components (clays), implicating another mechanism as well as improved carbonate preservation and increased coccolith export productivity affecting sedimentation at Site U1443. We suggest that an increase in coccolith $CaCO_3$ flux to the seafloor could have led to increased

scavenging by sinking biogenic aggregates of fine clays. Fine clays are present in the southern BOB water column as a direct result of riverine flux (Rixen et al., 2019; Ramaswamy, 1993), and in nepheloid layers above the NER where high clay concentrations occur due to proximity to the sedimentary fan systems to the east (Nicobar Fan) and west (Bengal Fan) (Stow et al., 1990). Recent studies of sedimentation patterns on the Bengal and Nicobar Fans, separated by the NER,

interpret a large increase in SAR both on the NER and the Nicobar Fan at ~10-8 Ma to reflect increased lithogenic sediment flux to the eastern Indian Ocean (Mcneill et al., 2017; Pickering et al., 2020b). Our data from the NER show that >75% of the 3-fold increase in SAR at 8.66 Ma is driven by biogenic $CaCO_3$, thus we caution against using SAR at Site U1443/758 as representative of changes in sediment flux to the Bengal-Nicobar Fan system. Data from Site U1443, as well as from

nearby DSDP Site 216 (Bukry, 1974; Mcneill et al., 2017; Pimm, 1974) (Fig. 1c), suggest that increases in biogenic carbonate accumulation on the NER are decoupled, both temporally and mechanistically, from the increase in sediment delivery to the Nicobar Fan system. The gradual increase in terrigenous element and non-$CaCO_3$ MARs over the 9-5 Ma interval seen at Site U1443 (Fig. 7g,i) is part of a longer-term trend of increasing mineral flux in this region of the NER from

the Miocene to the Pleistocene, beginning at ~12 Ma at ODP Site 758, that is thought to reflect increased Himalayan erosion (Ali et al., 2021; Hovan and Rea, 1992).

Increases in the MAR of biogenic components ($CaCO_3$, opal, organic carbon, phosphorus) between ~9 and 4 Ma have been measured in sediments from the Pacific, Indian and Atlantic Oceans (Farrell

et al., 1995; Lyle and Baldauf, 2015; Van Andel et al., 1975; Grant and Dickens, 2002; Delaney and Filippelli, 1994; Hermoyian and Owen, 2001; Dickens and Owen, 1999; Drury et al., 2020). This period of increased biogenic sedimentation, supported by independent paleoproductivity proxies (e.g., Diester Haass et al., 2005), is thought to reflect higher biological productivity and was dubbed the "biogenic bloom" by Farrell et al. (1995). A low-resolution $CaCO_3$ MAR record from Site 758

shows higher values between 8 and 4 Ma, which in the absence of evidence for an increase in carbonate dissolution at 4 Ma, could suggest an end to the biogenic bloom at this site in the early



Pliocene (Dickens and Owen, 1999; Pierce et al., 1989; Si and Rosenthal, 2019), although improved age control for the Pliocene interval as well as independent paleoproductivity reconstructions are needed to verify this. Hypotheses to explain the biogenic bloom invoke a change in global nutrient

cycling; i.e., a global increase in nutrient input, and/or redistribution of nutrients between basins (Grant and Dickens, 2002), although the asynchronous timing of the biogenic bloom between regions, its variable expression, and its differentiation from the carbonate crash recovery complicate its interpretation. Diester Haass et al. (2006) hypothesised that changes in reconstructed productivity were correlated to the LMCIS at four Indo-Pacific sites, and tentatively proposed a link

to a strengthened wind regime at this time. At Site U1443, we find no clear link between export productivity or carbonate sedimentation and the LMCIS (Fig. 7). In the northern Indian Ocean, the influence of possible concurrent changes in monsoon strength on paleoproductivity and biogenic MARs must also be considered, and these are discussed in Section 4.3.

**5.2 Orbital forcing of late Miocene South Asian summer monsoon winds**

On orbital timescales, time series analyses reveal dominant precession-band (22-24 kyr) variance in late Miocene export productivity records (Fig. 8, 9). Spectral analyses of individual calibrated timeseries of [Ba], [Al], [Fe], [Ti], $[Ba]_{cfb}$ and $\%CaCO_3$ allow us to tease apart the effects of sediment dilution and the competing influence of $Ba_{terr}$ and $Ba_{bio}$ on our Ba proxies, $[Ba]_{xs}$ and

log(Ba/Fe). The 41 kyr obliquity period is most significant (>90%) in the terrigenous element records (Al, Fe, Ti, Ba), and absent or less significant in $[Ba]_{xs}$, log(Ba/Fe), log(Ca/Terr), and $\%CaCO_3$ (Fig 8). The 24 kyr period stands out as highly significant in all records (>99%, except for $\%CaCO_3$ and $[Ba]_{cfb}$ where >95%). In contrast, the 125-kyr and 22.5 kyr peaks that dominates the $[Ba]_{xs}$ and log(Ba/Fe) spectra (>99% significant), are also highly significant only in the [Ba],

$[Ba]_{cfb}$, $\%CaCO_3$, and log(Ca/Terr) records. This suggests that strong variability at the 125-kyr (eccentricity) and 22.5 kyr (precession) periods is related to biological productivity (i.e. $Ba_{bio}$ and not $Ba_{detrital}$, as well as biogenic $CaCO_3$). The 30-kyr peak in $[Ba]_{xs}$ is also seen in [Ba] but not in [Fe], [Al], or [Ti], so we similarly interpret this period as being related to biological productivity. The 23.6-kyr and 22.3-kyr periods (highlighted together in Fig. 8 as one grey band spanning 22-24

kyr) are primary periods of Earth's precession, whereas the 53-kyr and 41-kyr periods are related to Earth's obliquity (Laskar et al., 2004).

Significant variability at obliquity and precession periods has been identified in high-resolution late Miocene-early Pliocene records of precipitation/runoff based on planktic foraminiferal $\delta^{18}O$ and



seawater $\delta^{18}$O in the nearby Andaman Sea (Jöhnck et al., 2020). These authors suggest that, prior to

a distinct switch to obliquity-driven variability around 5.55 Ma, their records reflect strong

precessional (insolation) control on South Asian monsoon rainfall from 6.2-5.55 Ma, with

significant phase lags between proxies and precession. Wavelet analyses of log(Ba/Fe) and [Ba]$_{xs}$

show that precession-band (22-24 kyr) variability dominated throughout our 9-5 Ma study interval

at Site U1443 (Fig. 9). Although phase relationships with insolation should be interpreted with

caution because of errors inherent to our late Miocene age model, export productivity appears to be

coherent and in phase (within error) with the summer inter-tropical insolation gradient (SITIG, the

insolation gradient between 23°N and 23°S on June 21$^{st}$) (Fig. S3d). The SITIG has been proposed

as a primary control on the strength of SASM winds, because a stronger SITIG increases the

pressure gradient between the two limbs of the winter hemisphere Hadley cell, which drives

monsoon winds into the summer hemisphere (Bosmans et al., 2015). Our new Ba-based export

productivity records corroborate the hypothesis that insolation played a dominant role in driving

late Miocene South Asian summer monsoon wind variability in the equatorial sector of the SAM

region, as predicted by general circulation models (Bosmans et al., 2018), with internal climate

processes such as ice volume playing a more minor role than in the Late Pleistocene when large

glacial-interglacial cycles and related feedbacks drove variability in the Asian monsoon on 100-kyr

timescales (Clemens et al., 2018; Clemens et al., 2021) and SASM wind proxies from the Arabian

Sea and southern BOB record up to ~9 kyr phase lags relative to precession (Bolton et al., 2013;

Caley et al., 2011; Clemens and Prell, 2003; Clemens et al., 1991).


In addition to the periods discussed above, a number of non-primary orbital periods termed

heterodynes, which result from non-linear interactions between variables operating at Earth's

primary orbital periods (Rial and Anaclerio, 2000; Thomas et al., 2016; Clemens et al., 2010), stand

out in our late Miocene records (24, 26, 30, 37, 49, and 68 kyr periods; Fig. 8). For example, the

1/24 kyr heterodyne, prominent in all our records, could result from the interference of eccentricity

with precession, and the 1/30 kyr heterodyne seen in Ba records from an interaction between

obliquity and precession. Several of these heterodynes have been previously identified in spectra of

seawater $\delta^{18}$O that reflect Asian monsoon precipitation and runoff, both in the Andaman Sea (30

and 130 kyr during the Pleistocene, 27 and 30 kyr in the latest Miocene) (Gebregiorgis et al., 2018;

Jöhnck et al., 2020) and in the East China Sea (29 and 69 kyr during the Pleistocene) (Clemens et

al., 2018), interpreted to suggest a strongly non-linear response of the monsoon to orbital forcing.

We favour the interpretation that the strong 24-kyr variability in our records reflects a primary

period of precession, because precession filters of [Ba]$_{xs}$ and log(Ba/Fe) spanning 22-25 kyr show





strong amplitude modulation of the precession signal at a period of ~405 kyr, which results from the

interaction of the 23.6-kyr and 22.3-kyr periods $(1/[(1/22.3) – (1/23.6)] = 404.8 \text{ kyr})$ (Fig. 10).
Amplitude modulation of precession-scale variability in our productivity records broadly follows
that of the SITIG (Fig. 10), suggesting a direct response of SASM winds to cross-equatorial
insolation gradients during the late Miocene.

Cross-spectral analysis of our $[Ba]_{xs}$ productivity record with the Site U1448 seawater $\delta^{18}O$ record
over the interval 4.95-6.19 Ma (where records overlap) shows >80% coherency and an in-phase
relationship at the 30-kyr period, suggesting that during this time monsoon winds and
precipitation/runoff in the BOB were to some degree coupled on orbital timescales (Fig. S3c), as is
the case in the late Pleistocene (Clemens et al., 2021). Nevertheless, our $[Ba]_{xs}$ record over this time

window contains stronger primary precession (22-24 kyr) and obliquity (41 kyr) signals than the
Site U1448 seawater $\delta^{18}O$ record that cannot be explained by differences in resolution, highlighting
that different climatic processes and feedbacks operating on orbital timescales must contribute to
the two records (interpreted to reflect runoff/precipitation and wind, respectively) to different
extents. In the late Pleistocene, strong obliquity-band and precession-band variance is found in

Andaman Sea records of monsoon precipitation/runoff (Gebregiorgis et al., 2018), whereas records
of upper ocean stratification controlled by South Asian monsoon wind mixing at Site 758 (~100 m
from Site U1443) show only precession-band variance (Bolton et al., 2013). The significant 41-kyr
variability seen in late Miocene terrigenous elements at Site U1443 (Fig. 8a-c) could also suggest
obliquity control on monsoon runoff into the BOB at this time, inferring a partial decoupling

between monsoon winds (controlling open-ocean productivity) and runoff (controlling terrigenous
sedimentation and salinity/seawater $\delta^{18}O$) on orbital timescales, although additional records from
regions closer to river sediment and freshwater sources are needed to corroborate this idea. It is
important to note that whilst SAM expression above Site U1443 in the southern BOB is dominated
by summer monsoon winds driving surface ocean currents and deeper mixing, oceanographic

conditions in the northern and eastern BOB (e.g., Sites U1447 and U1448) are instead primarily
controlled by summer monsoon freshwater inputs (Jöhnck et al., 2020; Kuhnt et al., 2020). Runoff
and direct precipitation during the SASM lead to strong salinity stratification in the northern parts
of the BOB in the late summer and autumn that prevents upper ocean mixing (e.g., Sengupta et al.,
2016). These regional differences in the manifestation of the monsoon must be considered when

interpreting records from the heterogeneous BOB, and records from multiple locations and proxies
are needed to achieve a representative picture of the SAM subsystem.



The 405 kyr eccentricity modulation of precession-scale export productivity variability broadly coincides with 405-kyr cycles in benthic $\delta^{13}$C at Site U1443, with higher export productivity occurring during benthic $\delta^{13}$C minima and eccentricity maxima on these timescales (Fig. 10). Cross-spectral analysis indicates that log(Ba/Fe) and benthic $\delta^{13}$C are > 95% coherent at the ~405 kyr and 22-24 kyr periods, with an in-phase relationship in the precession band (-13°±25) and a near antiphase relationship on 405-kyr timescales (151°±27) (Fig. S3b). 405-kyr modulation of the ocean carbon cycle, primarily recorded in carbonate content and benthic $\delta^{13}$C records (Herbert, 1997; Drury et al., 2020; De Vleeschouwer et al., 2020; Westerhold et al., 2020; Pälike et al., 2012; Paillard, 2017; Holbourn et al., 2007) but also in productivity and monsoon-related dust records (Rickaby et al., 2007; Wang et al., 2010), has been observed throughout the Cenozoic and Mesozoic sedimentary record. In middle Miocene records, poor carbonate preservation noted during eccentricity maxima is interpreted as indicating transient shoaling of the carbonate compensation depth (Holbourn et al., 2007; Flower and Kennett, 1994). Here, we see a broad positive correlation between log(Ba/Fe) and log(Ca/Terr) records on 405-kyr timescales (Figs. 9a,b), suggesting that carbonate content fluctuations at Site U1443 in the late Miocene were more strongly related to biogenic production than to dissolution on long eccentricity timescales. The coincidence of late Miocene eccentricity maxima with productivity maxima and benthic $\delta^{13}$C minima at Site U1443 is compatible with the hypothesis that during eccentricity maxima, a strengthened monsoon induced 405-kyr cycles in the marine carbon cycle via increased weathering and nutrient inputs, leading to enhanced marine biological productivity and deep-ocean organic carbon burial (Ma et al., 2011).

### 5.3 Late Miocene monsoon evolution

On long timescales, our Site U1443 biogenic Ba records show relatively stable (9 to 6.5 Ma) or slightly decreasing (6.5-5.3 Ma) export productivity between 9 and 5 Ma (Fig. 7d-f, Fig. S7). Based on sediment colour properties and XRF-derived Ba/Ti ratios in the preceding interval (~13.5-8.3 Ma), Lübbers et al. (2019) suggested that a shift towards a higher productivity regime at Site U1443 occurred at ~11.2 Ma, significantly earlier than the onset of the biogenic bloom at other sites and 2.5 Ma before the rise in CaCO$_3$ MAR, and that this shift was potentially linked to an intensification of the South Asian monsoon. An increase in export productivity at ~11 Ma is coherent with long-term changes in benthic foraminiferal assemblages at Site 758 (Gupta et al., 2004; Nomura, 1995) and at sites in the western tropical Indian Ocean (Smart et al., 2007). Reconstructions of Arabian Sea upwelling, export productivity, and deoxygenation (Bialik et al., 2020; Gupta et al., 2015; Huang et al., 2007; Zhuang et al., 2017), as well as the abrupt appearance of drift sediments in the



Maldives Archipelago at ~13 Ma (Betzler et al., 2016), point towards an intensification of seasonally-reversing South Asian monsoon winds between 13-11 Ma, consistent with Site U1443 export productivity records. Our new data suggest that similar levels of export productivity to those seen from 11.2 Ma to 9 Ma persisted until at least 5 Ma at Site U1443.


Compiled Asian monsoon proxy records spanning the 9 to 5 Ma interval show relatively stable long-term SASM strength (Fig. 11, see Fig. 1c for site map). In line with records from the Maldives and Arabian Sea (Tripathi et al., 2017; Huang et al., 2007; Betzler et al., 2016; Zhuang et al., 2017), Site U1443 records do not corroborate the hypothesis that SASM winds intensified at ~7-8 Ma as

suggested by some studies (Kroon et al., 1991; Singh and Gupta, 2014; Gupta et al., 2015; An et al., 2001). A recent modelling study suggests that the emergence of the Arabian Peninsula played a key role in the establishment of modern Somali Jet structure above the western Indian Ocean, initiating strong upwelling along the Oman margin during the Miocene (Sarr et al., in review). This provides a potential explanation for the temporal decoupling between western Arabian Sea records and

SASM records from other regions, and suggests that Arabian Sea upwelling records should not be interpreted as exclusive recorders of SASM strength on geological timescales. The long-term trend in our record shows broad agreement with a low-resolution clay mineralogy record from Site U1447 in the Andaman Sea showing gradual long-term decrease in smectite/(illite and chlorite) over the late Miocene (Fig. 11g), indicating strengthened physical weathering and/or weakened

chemical weathering, attributed to the South Asian winter and summer monsoons respectively (Lee et al., 2020). Also at Site U1447, records of potassium content (%K, Fig. 11x) are interpreted to show a shift in sediment provenance and/or an increase in physical weathering and erosion in the sediment source region between ~7 and ~6 Ma (Fig. 11h), potentially linked to an increase in monsoon rainfall intensity and global cooling (Kuhnt et al., 2020). Between 6.2 and 5 Ma, our

equatorial Indian Ocean wind records show good long-term agreement with a seawater $\delta^{18}$O record from the Andaman Sea (Jöhnck et al., 2020), with a minimum in export productivity at ~5.3 Ma at Site U1443 coinciding with a maximum in seawater $\delta^{18}$O at Site U1448 (Fig. 11i, j). One interpretation of this could be a coupled reduction in both SASM wind intensity and runoff/precipitation over this interval, although Jöhnck et al. (2020) invoke an increase in local

evaporation and/or a change in precipitation source to explain the decreasing seawater $\delta^{18}$O values between 5.6 and 5.2 Ma. High-resolution records of precipitation and runoff from the SASM region further back in time are needed to verify to what extent monsoon winds and precipitation/runoff are coupled on long timescales.



In contrast to the relatively stable South Asian monsoon between 9 and 5 Ma, evidence for a step
strengthening of the East Asian winter monsoon during the late Miocene (~7 Ma) comes from the
South China Sea (Holbourn et al., 2018) (Fig. 11a), whereas records from a site on the Chinese
Loess Plateau suggest a more gradual intensification of the East Asian summer monsoon from 8.2
to 2.6 Ma (Fig. 11b). The step change in South China Sea surface water geochemistry is interpreted

to reflect drying and cooling of the Asian interior and a related southward shift of the ITCZ leading
to an intensified dry winter monsoon over southeast Asia (Holbourn et al., 2018). The long-term
increase in East Asian summer monsoon strength inferred from magnetic records in Chinese loess
sequences is attributed to progressive Antarctic glaciation that drives an increased pressure gradient
between the Australian High and Asian Low pressure cells, a mechanism supported by numerical

simulations (Ao et al., 2016). However, we note that these single site records may not be
representative of the East Asian monsoon subsystem as a whole. Conversely, the apparent
insensitivity of equatorial SASM wind intensity to global late Miocene sea surface cooling, which
began at ~7.5 Ma and culminated in an SST minimum at 6 to 5.5 Ma (Fig. 11k) (Herbert et al.,
2016), is consistent with climate modelling studies that show limited impact of different $p$CO$_2$

scenarios on SASM wind patterns and strength (Kitoh et al., 1997; Sarr et al., in review). The South
Asian monsoon is widely considered to be a thermally direct circulation, driven by the
thermodynamic contrast between the Indian subcontinent and the equatorial Indian Ocean that
develops in summer, with changes in this gradient impacting the strength of onshore SASM
monsoon flow (Lutsko et al., 2019; Acosta and Huber, 2020). The lack of a long-term trend in wind

and surface circulation proxies over the 9 to 5 Ma interval (Fig. 11) suggests a relatively constant
land-sea temperature gradient despite global cooling. Thus, our data add to the body of evidence
suggesting decoupling between the East Asian and South Asian monsoons on long timescales.

## 6 Summary & Conclusions

We present new equatorial Indian Ocean benthic δ$^{13}$C and δ$^{18}$O records and an age model spanning
the interval between 9 and 5 Ma (late Miocene-earliest Pliocene), and analyse sedimentation and
productivity trends and cyclicity using XRF-derived records and MARs. Biogenic sediment MARs
reveal a modest imprint of the late Miocene biogenic bloom at Site U1443 lasting until at least 5
Ma, primarily driven by fine-fraction (coccolith) CaCO$_3$ accumulation, as noted at other sites (Si

and Rosenthal, 2019). Nevertheless, the carbonate MAR record of Site U1443 is clearly influenced
by carbonate preservation as well as production over the late Miocene, so independent productivity
proxies must be considered when defining the duration of the biogenic bloom. Our results indicate



that step increases in SAR recorded on the Ninetyeast Ridge at ~8-9 Ma primarily reflect an
increase in $CaCO_3$ accumulation, and that this is likely independent from the increase in lithogenic
sediment flux recorded in nearby Nicobar Fan sites, itself related to sediment re-routing within the
Nicobar-Bengal Fan system around the same time (Mcneill et al., 2017; Pickering et al., 2020a).
Our data show no long-term increase in export productivity between 9 and 5 Ma (and by inference
no intensification of SASM winds), therefore these data support existing evidence for an early late
Miocene (~13-10 Ma) establishment of strong seasonally reversing South Asian monsoon winds
and Arabian Sea upwelling, with relatively stable or slightly weakening SASM winds over the
remainder of the late Miocene and earliest Pliocene between 9 and 5 Ma. Spectral and cross-
spectral analyses of XRF-based biogenic barium records reveal that export productivity in waters
overlying Site U1443 was consistently paced by precession, with amplitude modulation of the
precession signal on ~405 kyr timescales and no significant variability at glacial-interglacial
(obliquity) timescales. Coeval late Miocene productivity maxima and benthic $\delta^{13}C$ minima during
eccentricity maxima at Site U1443 provides support for the hypothesis that the monsoon may have
paced changes in the carbon cycle on ~405 kyr timescales (Ma et al., 2011). Significant coherence
and an in-phase relationship at the precession band between biogenic barium and the SITIG
suggests direct forcing of South Asian monsoon winds by insolation gradients over the late
Miocene, relatively unaffected by glacial boundary conditions and long-term global cooling trends
(Herbert et al., 2016). In contrast, East Asian summer and winter monsoons appear to have
intensified during the late Miocene in response to global cooling and Antarctic ice sheet growth and
related feedbacks (Ao et al., 2016; Holbourn et al., 2018), although more continuous records over
the late Miocene are needed to understand regional trends due to the heterogeneous nature of Asian
monsoon expression.

## Data Availability

All data will be available on the www.pangaea.de database (submitted 18.06.2021, awaiting
validation)


## Supplement Link

## Author contributions

C.T.B. designed the study. Sample processing and picking of benthic foraminifera was carried out
by E.G and C.T.B. Picked foraminifera samples were verified and cleaned by A.H., W.K., and J.L.,





and stable isotopes measurements were performed by N.A., A.H., W.K., and J.L. XRF scanning

was carried out by C.T.B. in collaboration with K.G., G.M., and E.J.R. XRF calibration was carried

out by E.G. and K.T. The manuscript was written by C.T.B with feedback from all authors.

## Competing Interests

The authors declare that they have no conflict of interest.

## Acknowledgements

This research used samples and data provided by the International Ocean Discovery Program

(IODP). We thank the science party, technical staff and crew of IODP Expedition 353. Funding for

this research was provided by French ANR project iMonsoon ANR-16-CE01-0004-01 (CTB),

IODP France (CTB), and Deutsche Forschungsgemeinschaft grant Ku649/36-1(WK). KMG is

supported by Australian Research Council grant DE190100042. CTB thanks Anna Joy Drury and

Tim Herbert for age model feedback, and Luc Beaufort, Baptiste Suchéras-Marx, and Ian Bailey for

discussions that helped improve the manuscript. Marta Garcia Molina, Jean-Charles Mazur, and

Christine Pailles are thanked for technical laboratory support at CEREGE.

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



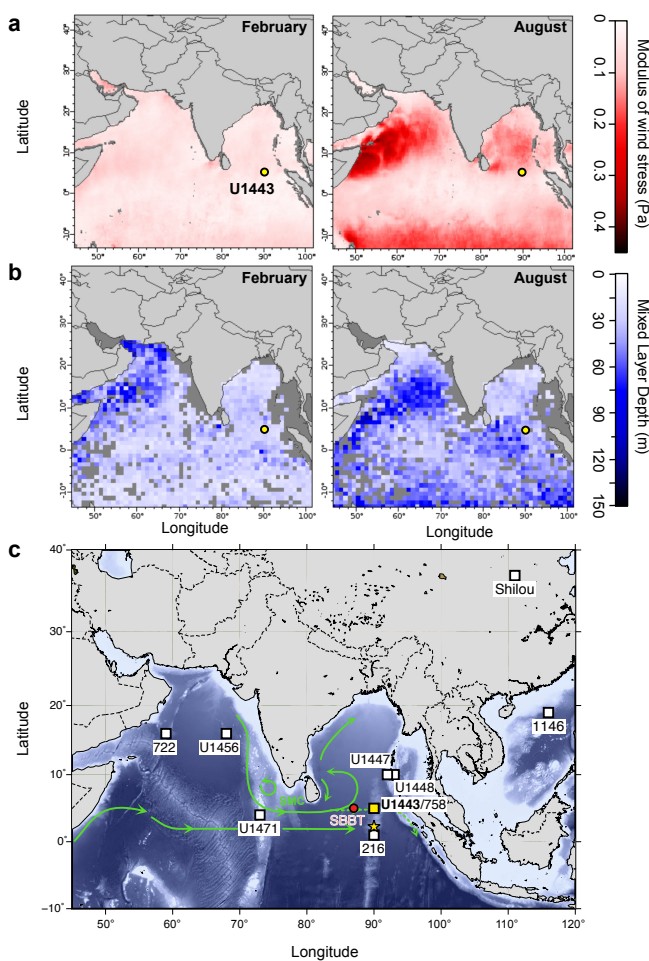

**Figure 1**: Seasonal contrast (February vs August) in wind stress (**a**) and mixed layer depth (**b**). Yellow dots indicate the modern location of IODP Site U1443. Maps were created on the ERDDAP website using the datasets *Wind Stress, Metop-A ASCAT, 0.25°, Global, Near Real Time, 2009-present (Monthly)* and *Ocean Climatology Ocean Mixed Layer Depth MLD T02 kriging* (see methods for details). **c**: Regional bathymetric map showing modern locations of marine and terrestrial sites discussed in this study (white squares) and the location of Site U1443, a redrill of Site 758, in the modern ocean (yellow square) and its paleolatitude at 10 Ma (yellow star, ~2°N, data from paleolatitude.org). The red circle shows the location of the SBBT sediment trap. Green arrows show surface ocean circulation during the summer monsoon (July/August) and the eastward flow of waters from the Arabian Sea into the BOB via the Southwest Monsoon Current (SMC), after Schott et al. (2009).


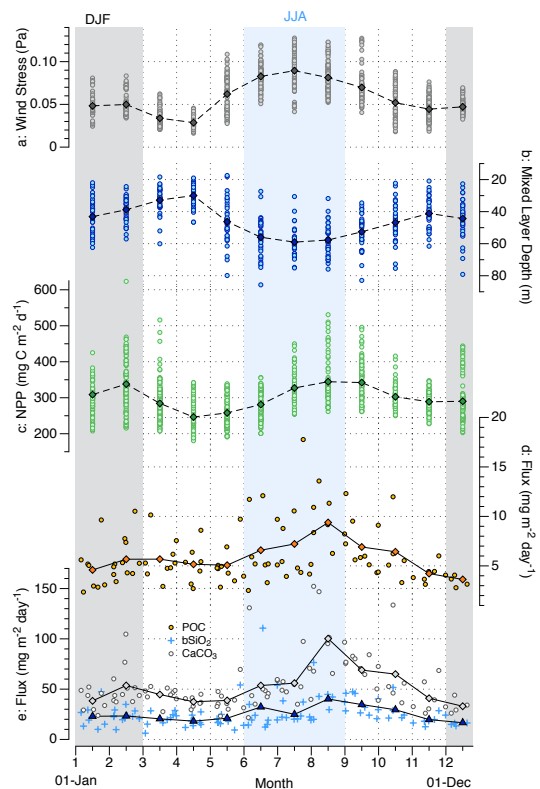

**Figure 2**: Modern, seasonal oceanographic variability above Site U1443 in the southern BOB. **a**: wind stress, **b**: mixed layer depth (MLD), **c**: net primary productivity (NPP), **d**: particulate organic carbon (POC) flux, **e**: biogenic silica (bSi) and calcium carbonate (CaCO3) flux. See Section 2.1 for
details of individual datasets and sources. Points represent individual months, diamonds and triangles with lines represent monthly mean values over entire time series. Months (x-axis) run from 1 (1st January) to 12 (1st December). JJA = June, July August, DJF = December, January, February.



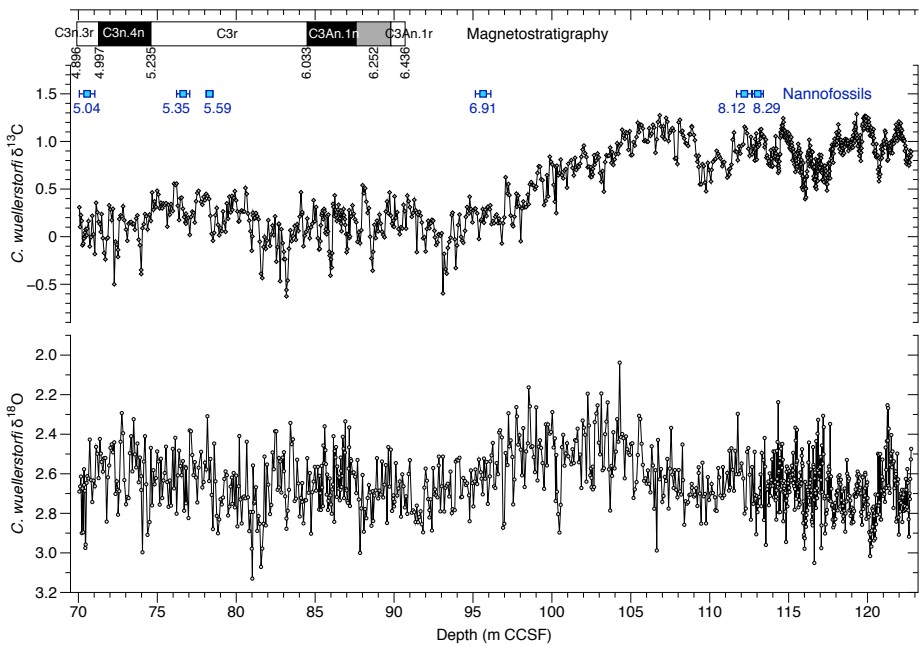


**Figure 3**: U1443 benthic foraminiferal (*Cibicidoides wuellerstorfi*) δ$^{18}$O (bottom) and δ$^{13}$C (top) data on the composite depth scale. Blue squares show refined depth ranges for calcareous nannofossil datums (see Table S1), and shipboard magnetostratigraphy is also shown for the interval over which it could be reliably determined. Between ~90 and 128m CCSF, sediments in

cores from all holes showed scattered directional signals during pass-through magnetic remanence measurement, which hindered any determination of polarity patterns in this interval across the whole site (Clemens et al., 2016). Black and white zones = normal and reversed polarity, respectively; grey zones = magnetic polarity not clearly determined. All numbers are age assignations for boundaries/nannofossil events in Ma (Gradstein et al., 2012).


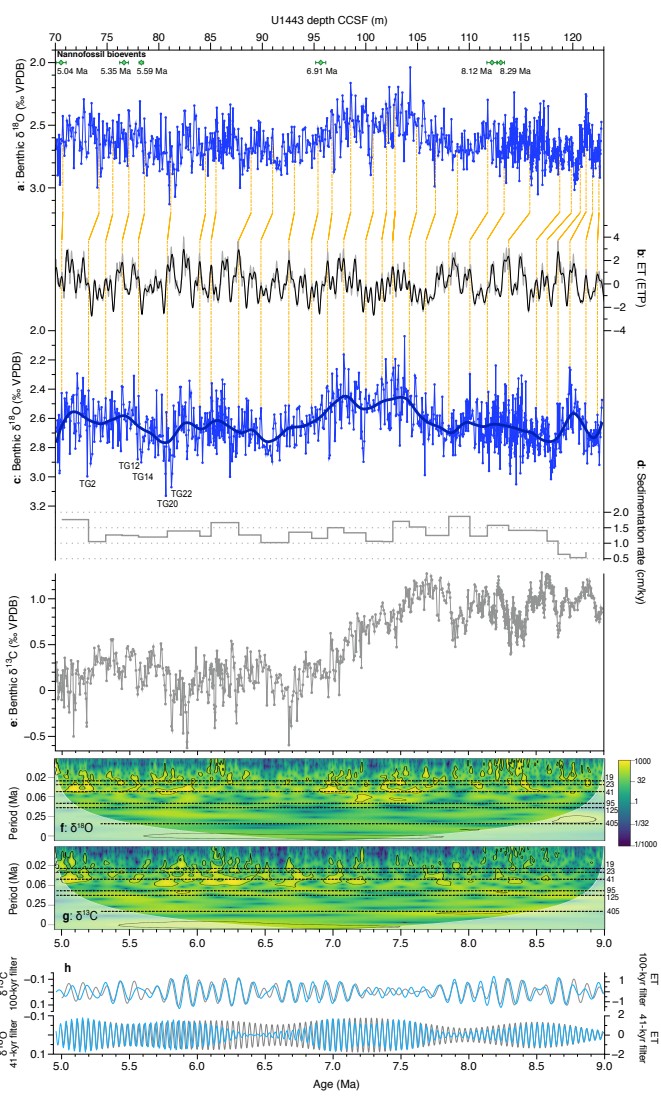

**Figure 4**: Astronomical (minimal) tuning of Site U1443 benthic δ18O record to ET target. **a**: benthic δ18O record on depth scale with nannofossil-based age constraints, **b**: ET tuning target (1:1 weighting, normalised). ETP (1:0.5:-0.4 weighting, normalised) is also shown in grey.
Astronomical time series from (Laskar et al., 2004). c: tuned benthic δ18O vs age, d: sedimentation rates, e: tuned benthic δ13C vs age. Tie-points between **a** and **b** are shown in orange (see Table 1). An age-depth plot showing ET tie-points and good agreement with biostratigraphic age control is shown in Fig S1. **f** and **g**: wavelet analyses of tuned isotope records; white shaded area shows cone of influence and contours show 95% significance level. Main orbital periods are shown on the right in kyr. **h**: Filtered tuned benthic isotope records compared to filtered ET (as in **b**). Top: 100-kyr
filtered benthic δ13C (blue; Gaussian filter centred on 100 kyr with bandwidth ±25 kyr to include 95 and 125 kyr peaks) compared with filtered ET (grey, identical filter design). Bottom: 41-kyr filtered



benthic $\delta^{18}$O (blue, Gaussian filter centred on 41.5 kyr with bandwidth ±1.5 kyr) compared with
filtered ET (grey, identical filter design).


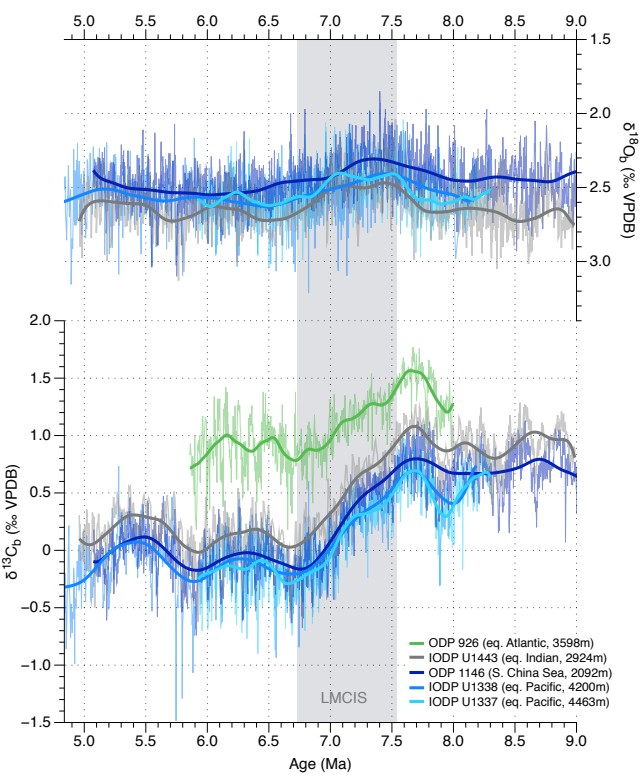

**Figure 5**: Late Miocene evolution of low-latitude deep-ocean inter-basin benthic $\delta^{13}$C and $\delta^{18}$O
gradients. South China Sea ODP Site 1146 (Holbourn et al., 2018) with age model revised in
(Holbourn et al., In Press), equatorial Pacific IODP Sites U1338 (Drury et al., 2018; Drury et al.,
2016) and U1337 (Drury et al., 2017), and ODP Site 926 (Shackleton and Hall, 1997; Drury et al.,
2017; Zeeden et al., 2013). All records are shown on their latest independent orbitally-tuned
chronologies. We have excluded Caribbean ODP Site 999 from this figure because it is bathed in
intermediate water masses due to basin geometry and sill depths (Bickert et al., 2004). Deep South
Atlantic Site 704 (Müller et al., 1991) data are not plotted due to clear age model discrepancies
when compared to orbitally-tuned records. All $\delta^{18}$O records are based on *Cibicidoides wuellerstorfi*
or *C. mundulus* therefore no corrections are applied, following (Jöhnch et al., 2021). The Site 926
record includes $\delta^{13}$C corrections for some samples due to the multispecific nature of the record
(Drury et al., 2017). No correction was applied to *C. wuellerstorfi* or *C. mundulus* $\delta^{13}$C values.




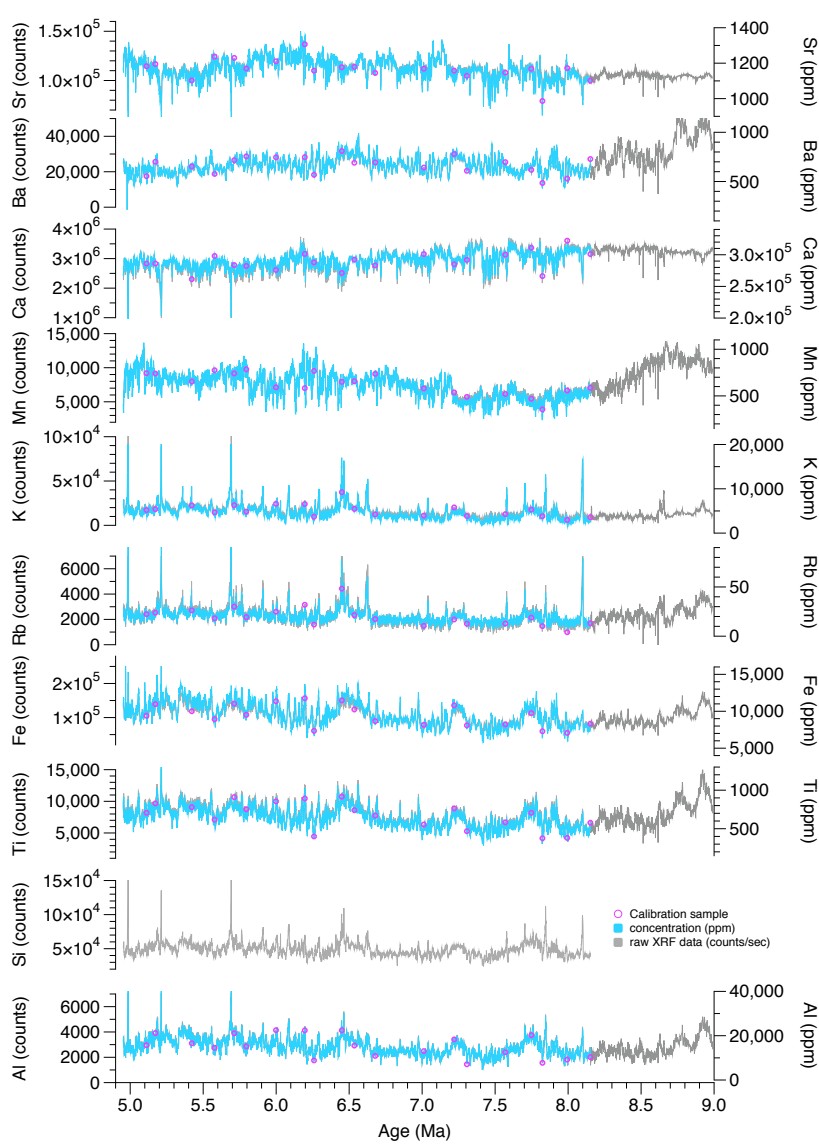

**Figure 6**: Raw counts/sec (grey lines) and calibrated concentration (blue lines) scanning XRF elemental data over the late Miocene interval. Pink circles show samples used for calibration (See Fig. S6).

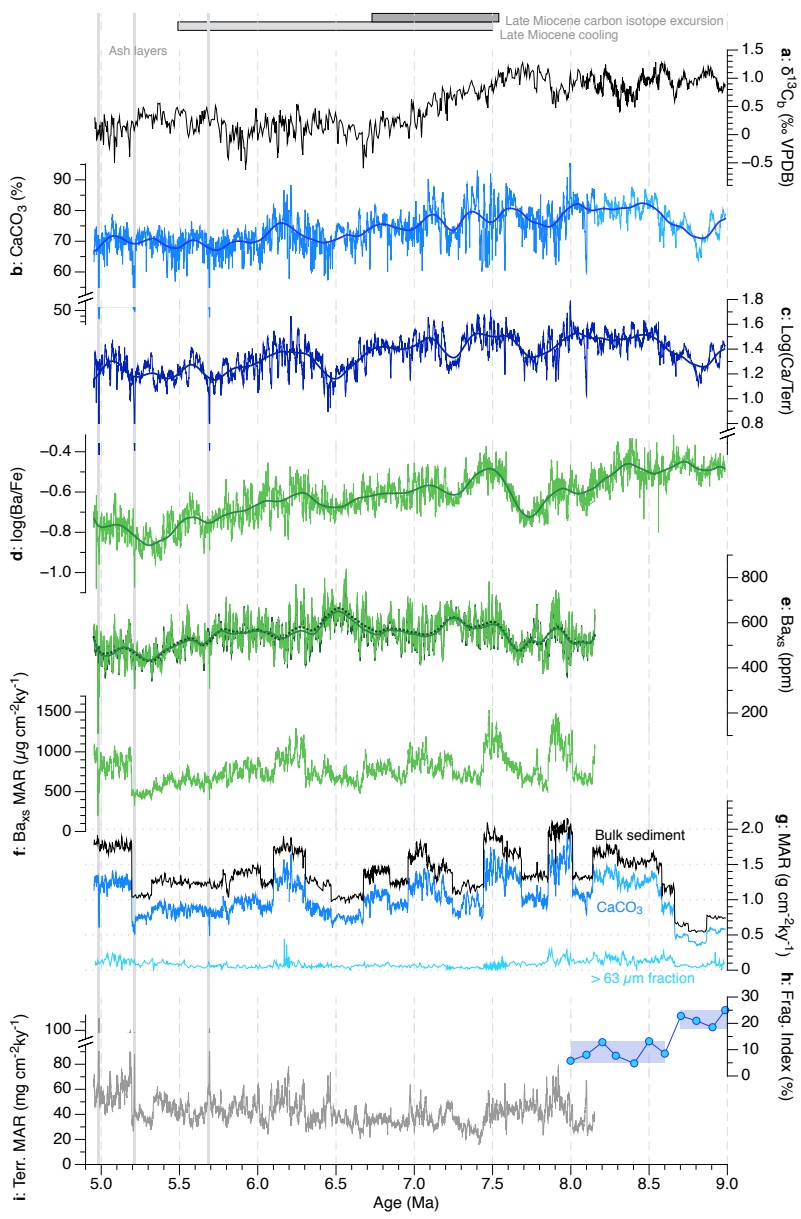


**Figure 7**: XRF-derived late Miocene CaCO₃ and export productivity records from Site U1443. **a**: benthic δ¹³C, **b**: percent CaCO₃, **c**: log(Ca/Terr), **d**: log(Ba/Fe), **e**: Ba$_{xs}$ calculated with both [Al] (light green) and [Ti] (dark green), **f**: [Ba]$_{xs}$ MAR, **g**: bulk (black), CaCO₃ (blue), and >63µm fraction (light blue) MAR, **h**: foraminiferal fragmentation index (following Le and Shackleton,

1992), **i**: Terrigenous (Al+Fe+Ti+K+Rb) MAR. For CaCO₃ records, lighter blue lines are based on % CaCO₃ from Lübbers et al. (2019) and darker blue lines are based on % CaCO₃ estimates in this study. The late Miocene carbon isotope excursion and the main interval of late Miocene SST cooling are shown as grey bars. Shaded intervals are identified ash layers.




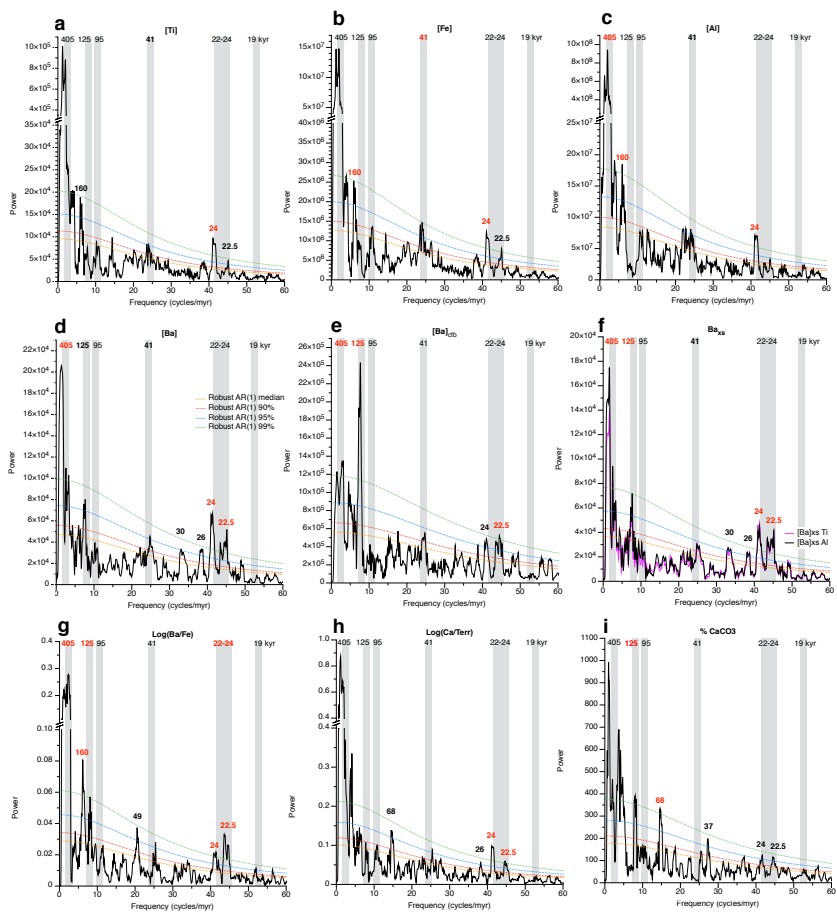

**Figure 8**: Spectral analyses for over the 9-5 Ma interval (log ratios and % CaCO₃) or 8.15-5 Ma interval (calibrated element concentrations and [Ba]ₓₛ). **a**: [Ti], **b**: [Fe], **c**: [Al], **d**: [Ba], **e**: [Ba]cfb (carbonate free basis), **f**: [Ba]ₓₛ, **g**: log(Ba/Fe), **h**: log(Ca/Terr), **i**: % CaCO₃. Grey bands denote primary orbital periods based on the La04 astronomical solution. The 22-24 kyr band covers two

peaks centred at 23.5 and 22.3 kyr.

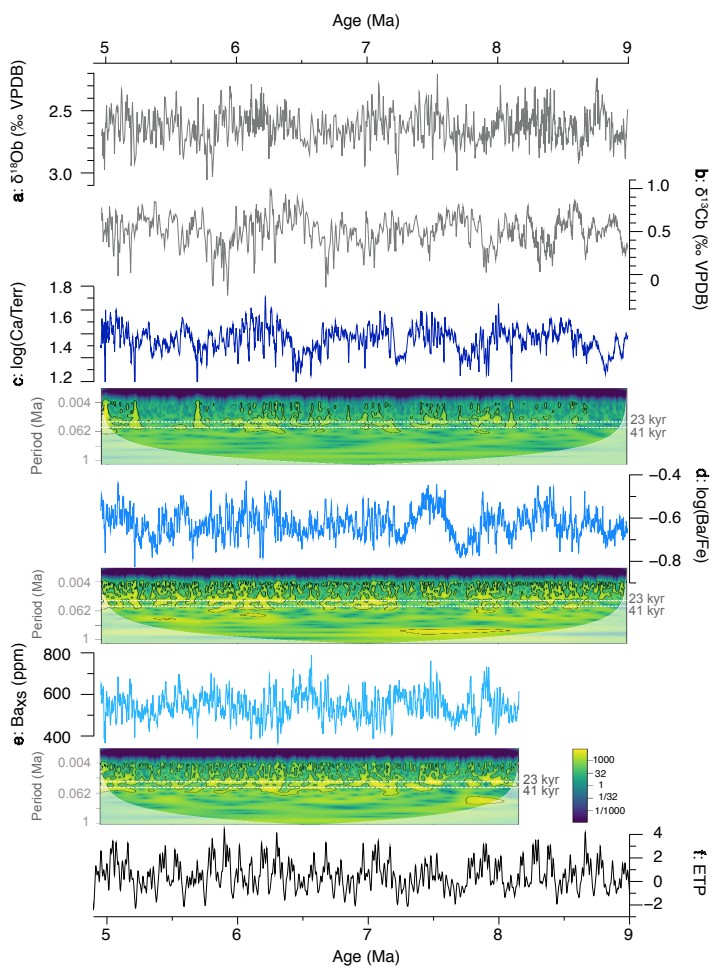

**Figure 9**: Orbital-scale variability of productivity and $CaCO_3$ proxies at Site U1443. All records
shown here are bandpassed as described in the methods. (a) benthic $\delta^{18}O$, (b) benthic $\delta^{13}C$, (c)
log(Ca/Terr), (d) log(Ba/Fe), (e) $[Ba]_{xs}$, (f) ETP. For c-e, wavelet analyses are shown, illustrating
dominant precession-scale (22-24 kyr) variability in Ba proxies and both precession and obliquity
(41-kyr) variability in log(Ca/Terr).


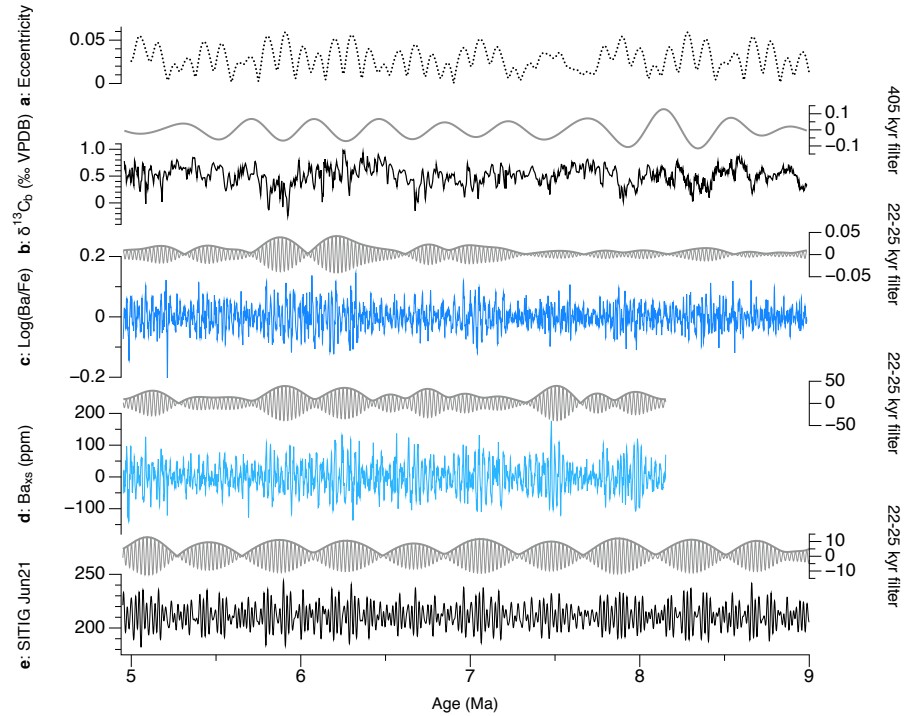

**Figure 10**: Detrended and filtered late Miocene U1443 records to illustrate precession-band variance and amplitude modulation. **a**: eccentricity (Laskar et al., 2004), **b**: benthic $\delta^{13}$C (bandpassed as in Fig. 9) and its 405-kyr filter (above) (note reversed y-axes). **c**: Lowess-detrended log(Ba/Fe) (window = 0.1 Ma) and 22-25 kyr filter (above), **d**: Lowess-detrended $Ba_{xs}$ (window = 0.1 Ma) and its 22-25 kyr filter (above), **e**: The summer inter-tropical insolation gradient (SITIG, calculated as the insolation difference between 23°N and 23°S on 21$^{st}$ June using orbital solution of Laskar et al. (2004)) and its 22-25 kyr filter and amplitude modulation (above); For panels **c-e**, raw datasets were filtered using a Tanner-Hilbert filter centred on 43 cycles/myr with bandwidth ±3 (designed to include both the ~24 kyr and ~22 kyr precession periods). For panel **b**, a Tanner-Hilbert filter centred on 2.47 cycles/myr with bandwidth ±0.8 was applied.

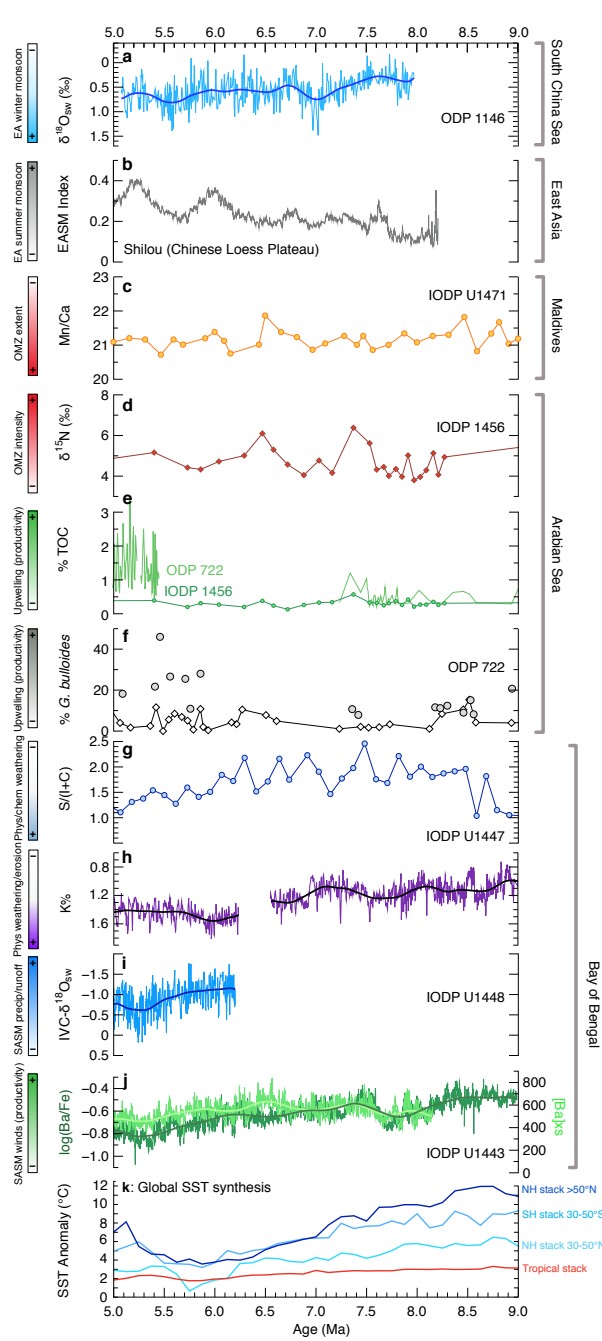

**Figure 11**: Compilation of late Miocene (9-5 Ma) Asian monsoon reconstructions, showing
representative records from different regions and their published interpretations (bars on left). **a**:
Seawater δ¹⁸O showing an increase in East Asian winter monsoon strength at ~7.4 Ma (Holbourn et
al., 2018; Holbourn et al., In Press), **b**: Stacked magnetic records of the East Asian summer





monsoon from the Chinese Loess Plateau (Ao et al., 2016), **c**: Mn/Ca ratios used to trace oxygen
minimum zone (OMZ) variations from Maldives Site IODP U1471 (Betzler et al., 2016), **d**: δ15N
record from Arabian Sea site IODP U1456 showing OMZ intensity (Tripathi et al., 2017), **e**: total
organic carbon (TOC) % from Arabian Sea sites IODP U1456 and ODP Site 722 (Huang et al.,
2007; Tripathi et al., 2017), **f**: % *Gobigerinoides bulloides*, a planktic foraminiferal upwelling
indicator, at ODP Site 722 (Huang et al., 2007; Kroon et al., 1991), *G. bulloides* was counted in the
>150 $\mu$m fraction in *Huang et al.* (white diamonds) and the >125 $\mu$m fraction in *Kroon et al.* (grey
circles), **g**: clay mineralogy (smectite/(illite + chlorite)) at IODP Site U1447 in the Andaman Sea
(Lee et al., 2020), **h**: Percentage Potassium (K%) at IODP Site U1447 derived from spectral natural
gamma ray measurements (Kuhnt et al., 2020), **i**: Andaman Sea IODP Site U1448 ice-volume-
corrected seawater δ18O record (Jöhnck et al., 2020), and **j**: Export productivity records from Site
U1443 (this study). **k**: Global sea surface temperature trends (expressed as anomalies relative to the
present), stacked by latitude band (Herbert et al., 2016). All records are on their original age
models, and Loess smooths are shown for high-resolution records.


