# Peer review of "Secular and orbital-scale variability of equatorial Indian Ocean summer monsoon winds during the late Miocene"

_Climate of the Past, 2021_

## Author Comment (AC1)

**RC1:**

"Secular and orbital-scale variability of equatorial Indian Ocean summer monsoon winds during the late Miocene" by Bolton et al. (CEREGE, Aix-en-Provence, France)

The manuscript by Bolton et al. presents new proxy records and an astronomically-tuned age-depth model from a recently-drilled IODP deep-ocean sediment core (U1443). Proxy records span the late Miocene (9 – 5 Ma) and include downcore benthic isotope records (d13C and d18O) and XRF-derived productivity-related and detrital-related elemental data. All proxy records are of sufficient resolution to resolve precession cycles, i.e. the shortest astronomical frequency. Based on their results, the authors present three important conclusions:

First, the authors observe a 3-fold increase in $CaCO_3$ mass accumulation rates at 8.66 Ma, but no change in their export productivity proxy log(Ba/Fe). They interpret this pattern as the result of a contemporaneous increase in coccolith productivity and improved preservation. This interpretation supports a weathering alkalinity and nutrient change as the driver for the expression of the so-called "biogenic bloom" in this region. Second, the authors infer that monsoonal dynamics throughout the studied interval are dominated by eccentricity-modulated precession on orbital timescale. Third, the authors do not find an intensification of the South Asian monsoon over the late Miocene, as has been proposed by some previous works.

The Site U1443 proxy records in themselves are precious and already deserve publication in their own right. The three conclusions that accompany them are an important step toward a mechanistic and regionally-differentiated understanding of late Miocene monsoon dynamics on orbital and geologic time scales. I thus recommend this paper for publication in Climate of the Past after minor revisions. Indeed, I would like the authors to consider my three major comments that could potentially make their paper even stronger.

We thank the reviewer for their really positive and constructive comments on our work.

**Major comments**

[1] Throughout the paper, the authors filter precession with a Tanner-Hilbert filter with a bandwidth between 40 and 46 cycles/Myr (22 – 25 kyr periodicities). This bandpass is too narrow to encompass all relevant precession components (see Table 1).

**Table 1.** Frequency decomposition of the precession of the Earth's axis, using g frequencies from Table 3 in Laskar et al. (2004) and the precession frequency of the Earth p = 50.475838 arcsec yr$^{-1}$. (p+g3) and (p+g4) are in red because they are important components of the precession frequency decomposition, yet they are not included in the used bandpass filter.

|        | "/year    | cycles/Myr  | kyr        | Planet     |
| ------ | --------- | ----------- | ---------- | ---------- |
| p+g1   | 56.065838 | 43.26067747 | 23.1156805 | Mercury    |
| p+g2   | 57.927838 | 44.69740586 | 22.372663  | Venus      |
| p+g3   | 67.843838 | 52.34864043 | 19.1026929 | Earth-Moon |
| p+g4   | 68.391838 | 52.77147994 | 18.9496296 | Mars       |
| p+g5   | 54.73329  | 42.23247685 | 23.6784597 | Jupiter    |
| p+g6   | 78.720838 | 60.74138735 | 16.4632394 | Saturn     |
| p+g7   | 53.563789 | 41.3300841  | 24.1954504 | Uranus     |
| p+g8   | 51.148859 | 39.46671219 | 25.3378086 | Neptune    |
| p+g9   | 50.125898 | 38.67739043 | 25.8548984 | Pluto      |

The inclusion of the (p+g3) and (p+g4) terms in a precession-centred bandpass filter is important for the correct amplitude demodulation. This is because the four most important terms that compose short eccentricity involve (p+g3) or (p+g4).

**Table 2.** Frequency decomposition of the four most important terms in the short eccentricity evolution of the Earth's orbit. These four frequencies all involve either (p+g3) or (p+g4). When these terms are not included in a precession-centred bandpass filter, the short eccentricity terms cannot be extracted from the filter's amplitude demodulation.

|               | "/year   | cycles/Myr  | kyr         |   |
|---------------|----------|-------------|-------------|---|
| (p+g3) - (p+g2) | 9.916    | 7.651234568 | 130.697862  |   |
| (p+g4) - (p+g2) | 10.464   | 8.074074074 | 123.853211  |   |
| (p+g3) - (p+g5) | 13.110548| 10.11616358 | 98.8517032  |   |
| (p+g4) - (p+g5) | 13.658548| 10.53900309 | 94.885635   |   |

The consequences of too-narrow precession filtering clearly appear in Figure 10. The amplitude modulation signals only exhibit low-frequency variations at the rhythm of the 405-kyr eccentricity cycle. The 405-kyr appears in the authors' amplitude demodulation because is created by (p+g2)-(p+g5) and both terms are included in the 22 – 25 kyr precession filter. The 100-kyr terms however do not appear because they require the inclusion of the (p+g3) and (p+g4) terms into the precession filter. I would thus strongly recommend the authors to widen their precession filtering settings. This will markedly improve the results since it can already be recognized by eye that there are ~100-kyr amplitude modulation cycles embedded in the Ba$_{xs}$ and log(Ba/Fe) time series (as well as in the SITIG forcing of course).

 We thank the reviewer for this useful comment, and for the very thorough explanation that accompanied it. The original decision to filter at 22-25 kyr was based on the presence of significant spectral peaks only within this band (and the absence of spectral peaks at ~19 kyr) in the MTM spectral analyses of Ba proxies, but we now see how this decision biased our results. We have widened the precession filter to include all of the relevant terms, and the new filter covers 18-26 kyr (frequency 46.5±8.5, 38-55 cycles/Myr). The 100-kyr amplitude modulation signal in now visible (as well as the 405 kyr one) in our filtered records in revised Fig. 10.

 [2] I find the obliquity peaks in the detrital proxies (Ti, Fe and Al) in Figure 8a-c intriguing. They do have about the same spectral power than the precession peaks. The authors briefly discuss the possibility that this result might indicate a decoupling between monsoon winds (driving productivity on precession timescales) and monsoon precipitation (terrigenous variability on obliquity timescales) [lines 617 – 621]. I would encourage the authors to explore this observation a little deeper. Does wavelet analysis show that obliquity primarily appears when eccentricity is low? Are there any modelling studies that corroborate this idea?

We also find the stronger 41-kyr variability in detrital proxies really interesting, and despite digging into the literature on this subject, we have yet to find a satisfactory explanation for the stronger obliquity signal in the runoff-related elements than in the wind-driven productivity signal.

Clemens *et al.* (2021) show that 100 kyr and 41 kyr variability are at least as important as precession in Pleistocene proxy records of monsoon precipitation/runoff in the Bay of Bengal, and suggest that summer monsoon precipitation is strongly influenced by global boundary conditions related to ice-volume and greenhouse gas feedbacks (which in the late Miocene, fluctuate on 41-kyr timescales). On the other hand, obliquity forcing of tropical climate has been shown to occur independently of high-latitude ice-sheet growth and decay as a result of interhemispheric insolation gradients (Bosmans et al., 2015). Yet changes in cross-equatorial moisture transport (and therefore monsoon precipitation) on precession and obliquity timescales related to the SITIG are expected to be coupled to changes in South Asian monsoon wind intensity, so this does not help reconcile the stronger obliquity signal in runoff relative to wind proxies at Site U1443 (although we note that a lower significance obliquity peak is visible in the [Ba]$_{xs}$ spectrum, and the cross-spectral analysis of the ~6.2-5 Ma interval of the [Ba]$_{xs}$ record with the Site U1448 seawater $\delta^{18}$O record shows significant obliquity – Fig. S3c).

A strong response to obliquity forcing was also recorded in late Miocene monsoonal runoff records in the eastern Bay of Bengal and was interpreted as related to changes in latitudinal and interhemispheric temperature gradients (Jöhnck et al., 2020). Model results show increased SE Asian summer monsoon precipitation and a northward shift of convection from ocean to land at minimum precession and maximum obliquity (Bosmans et al., 2018). The same set of fully coupled high-resolution models indicate a more complex and spatially heterogenous response of South Asian summer monsoon precipitation. In these models, wind speed is increased over the southern hemisphere tropical Indian Ocean for both precession and obliquity (Bosmans et al 2018), which likely is reflected in the orbital signature of the Site U1443 productivity records. A recent study using a coupled atmosphere–ocean general circulation model with emphasis on the relative roles of precession and obliquity changes also suggests that dynamic effects (changes in

winds) dominate the monsoonal response to both precession and obliquity forcing in most monsoonal systems (Ding et al., 2020).

Wavelet analyses for Fe, Al and Ti (see below, contours are 95% confidence intervals and pink line shows 41 kyr period) do not appear to indicate a correlation between strong obliquity variance and low eccentricity.

[Figure]

We have expanded the discussion of obliquity forcing of monsoon runoff in the revised manuscript, although the mechanisms behind the relatively stronger obliquity signal in runoff (terrigenous sedimentation) records compared to summer monsoon wind (export productivity) records at Site U1443 remain unexplained.

[3] The introduction nicely displays how there are two productivity peaks per year in the Bay of Bengal. This annual course creates the potential for half-precession cycles in the Barium-related productivity proxies. Indeed, one might expect productivity to be fuelled both during a precession minimum (stronger summer winds) and during a precession maximum (stronger winter winds). This potential is not discussed in the paper, yet the temporal resolution of the Ba proxies (<1 kyr) does allow the authors to report on the presence or absence of such cycles.

We agree with the reviewer that the absence of a semi-precession signal in our high-resolution Ba records, despite the near-equatorial location of our site and the double annual primary productivity peak in the modern ocean, is really interesting.

In the late Pleistocene at Site 758/U1443, a strong half-precession signal is detected in upper-water column stratification proxy records (Bolton et al 2013). Based on our interpretation of modern oceanographic data, we expect upper-ocean stratification and productivity to be coupled at this location, however we currently lack paleoproductivity data on these same Pleistocene samples to verify this (this is something we are working on).

One explanation for the lack of a half-precession signal in paleoproductivity proxies at this location might be related to the fact that export productivity (i.e. the fraction of net primary productivity that ends up accumulating in underlying sediments) is heavily biased towards the late summer monsoon season, perhaps as a result of increased ballasting by the higher concentration of biogenic particles and by terrigenous particles carried into the BOB by runoff. In Figure 2, although net primary productivity displays two clear peaks over the annual cycle, particle fluxes to deep sediment traps (~3000m) show a much smaller (CaCO3) or absent (particulate organic carbon and biogenic silica) peak associated with the winter monsoon. Thus, we think that the export productivity recorded in Site U1443 sediments represents first and foremost the summer monsoon (this is mentioned in Section 2). In the discussion (Section 5.2), we now explicitly discuss the lack of semi-precession signal in our records, and relate it back to the bias in particle export.

We also now note that the lack of a semi-precession signal in our records corroborates the idea that the SITIG (the summer inter-tropical insolation gradient), rather than local insolation (which contains a significant half-precession component between the equator and 5° latitude), was a primary driver of export productivity variations at our site.

**Minor Comments**

**Throughout:** A lot of acronyms are used. To my taste, a little too much. Please consider whether you could spell out some of them. For example: BOB, NER, SMC, MLD, NPP, …
We have removed the following acronyms from the main text to improve readability (some are still mentioned in figure captions only in relation to annotations): SMC, POC, NPP, SAR.

**Lines 60-65:** The use of X versus Y does not work well in all cases. I would recommend to spell out the contrast you would like the reader to consider.
We have tried to clarify this.

**Line 91:** Also check out Ding et al. (2021), Climate Dynamics 56
Thanks, we have added this reference.

**Line 136:** In … In … Delete repeated wording
corrected

**Line 177:** The geographic coordinates could be a little more precise.
Corrected (co-ordinates differ slightly for Holes A-D, we used Hole A co-ordinates)

**Line 185:** It is not exactly clear to me which splice has been used. There are two U1443 splices online on the IODP LIMS database, but both are already more than 5 years old. The authors should make the affine and splice tables available in the supplements, or on Pangaea, or cite a reference where the splice is available.
Sorry that this was not clear. We have added a table to the Supplementary File (new Table S1) listing splice intervals.

**Line 204:** avoid subjective qualifiers like "small"
This has been changed to 1° by 2° box.

**Line 236:** Replace "high-resolution" by "~1 meter resolution"
This has been changed to ~0.5-1m resolution

**Line 345:** What exactly is meant by "spectral analyses … on filtered records". Why would one do bandpass filtering prior to spectral analysis in this case?
What we mean here is that we carried out spectral analyses on records that had been detrended (filtered to remove signals with periods longer than one third of the length of the dataset (>1.6 Ma) using the "bandpass" function in Astrochron), so that long-term trends did not lead to a low-frequency period dominating the power spectra. We have clarified this in the text.

**Line 371:** The y-axes of the phase graphs are not very helpful, and even a bit misleading. Please cut them off at -180° and +180°. Of course, confidence intervals can go beyond this range, but it should be clear that -180° = +180° = anti-phased behaviour.
Thanks for this comment. We have changed the axes (and grid lines) on all cross-spectral phase plots so that they are limited at -180° and +180°, and have clarified that both 180 and -180° phases indicate anti-phased behaviour in the caption (Fig. S3).

**Lines 444 – 456:** I miss a statement here about the step-wise character of the MAR series. It should be acknowledged that these steps in MAR are related to age-model-induced stepped sedimentation rate changes.
We have added the following statement: "The stepwise nature of MAR records results from age model-imposed stepped changes in sedimentation rate."

**Line 544:** Section 5.3
Corrected

**Line 1259:** Section 3.1
Corrected

**References cited**

Bosmans, J.H.C., Hilgen, F.J., Tuenter, E. and Lourens, L.J., 2015. Obliquity forcing of low-latitude climate. *Climate of the Past*, *11*(10), pp.1335-1346.

Bosmans, J.H.C., Erb, M.P., Dolan, A.M., Drijfhout, S.S., Tuenter, E., Hilgen, F.J., Edge, D., Pope, J.O. and Lourens, L.J., 2018. Response of the Asian summer monsoons to idealized precession and obliquity forcing in a set of GCMs. *Quaternary Science Reviews*, *188*, pp.121-135.

Clemens, S.C., Yamamoto, M., Thirumalai, K., Giosan, L., Richey, J.N., Nilsson-Kerr, K., Rosenthal, Y., Anand, P. and McGrath, S.M., 2021. Remote and local drivers of Pleistocene South Asian summer monsoon precipitation: A test for future predictions. *Science Advances*, *7*(23).

Ding, Z., Huang, G., Liu, F., Wu, R. and Wang, P., 2021. Responses of global monsoon and seasonal cycle of precipitation to precession and obliquity forcing. *Climate Dynamics*, *56*(11), pp.3733-3747.

Jöhnck, J., Kuhnt, W., Holbourn, A. and Andersen, N., 2020. Variability of the Indian Monsoon in the Andaman Sea across the Miocene-Pliocene transition. *Paleoceanography and Paleoclimatology*, *35*(9), p.e2020PA003923.

---

## Author Comment (AC2)

**RC2**

Bolton et al use an XRF-derived, orbitally tuned Ba record from southern Bay of Bengal site U1443 in order to study changes in productivity and summer monsoon in the 9-5 Ma time period. They used XRF scanner barium to track productivity through time. They suggest that precessional variations were evidence of summer monsoon wind strength in the equatorial Indian ocean and that South Asian monsoon winds were established prior to 9 Ma, with no apparent intensification over the late Miocene. They have produced a data set that is worth publication.

My main concern that needs to be addressed is that during the period that they study, the site moved northward perhaps by 200 km (2° of latitude). They did not address how that movement may have affected the records they discuss and that needs to be considered. For the most part the data they have seems to agree that the late Miocene between 8 and 5 Ma are part of a global high productivity interval, and they don't observe evidence for intensification of the South Asian Summer Monsoon toward the present. The data are from Site U1443 from IODP Expedition 353 located near ODP Site 758 on the Ninety East Ridge. They have a good discussion of modern oceanography and its relationship both to winds and productivity. The study adds an important data set in a region that needs more records.

We thank the reviewer for their useful comments on our paper. We reply in detail to the concern on paleolatitude below.

Their description of the sediment column is in the Materials and Methods section needs more work. The depths of the late Miocene interval are not needed, and core-section-interval designations just clutter up the writing here, especially since specific sections from different holes are not discussed later. Why are they describing sampling for micropaleontology? I also didn't see CCSF depths for the interval they discuss.

We provide both CCSF depths and hole-core-section-interval information for our study interval in the spirit of making our study fully reproducible to those who may wish to test our ideas. Additionally, we now include a reference to a splice table in the supplement, which was added at the request of Reviewer 1. We have deleted the sentence detailing 1 cm sampling for micropaleontology – we originally included this to explain that we took 1-cm whole round samples for foraminiferal work (rather that 2-cm quarter-round samples as is common) because of low sedimentation rates.

In addition, figures in the Proceedings chapter on Site U1443, there seems to be a speed up of sedimentation immediately older than their interval. At what age did that happen? In figure 4 there is significantly lower sedimentation rates at the beginning of the 9-5 Ma interval—could these be the end of the lower sedimentation rate interval?

The late Miocene increase in sedimentation rate identified by shipboard bio-magnetostratigraphy at ~9 Ma (between 100-130m CSF-A, Fig. F14 in the U1443 Site Chapter) does indeed correspond to the increase in sedimentation rate at ~8.6 Ma in our new age model (Fig. 4d). This is confirmed by revised nannofossil biostratigraphy performed as part of this study (Fig. S1).

Given that the Site report gives the paleoposition of Site U1443 as 5°S at the Oligocene-Miocene boundary, what were its paleopositions during the 9-5 Ma time interval? A quick estimate shows that the site would have been between 1.5 and 3.2°N. Could this affect their interpretation?

The paleolatitude of Site U1443 at 10 Ma was ~2°N (based on paleolatitude.org), and this was shown on Figure 1c in the original submission (yellow star on map). We have now calculated more precise paleo-positions to ensure that our interpretations remain valid.

Based on the G-Plates online portal which allows calculation of paleo-positions at 1 Ma resolution (http://portal.gplates.org/service/reconstruct_points), the position of Site U1443 changed from **1.71°N, 88.06°E** at 9 Ma to **3.27°N, 89.04°E** at 5 Ma (see Fig. R1 below).

[Figure]

**Fig. R1: Paleo-position of Site U1443 (red dot) on paleogeographic maps for 9 Ma, 5 Ma and 0 Ma (computed using G-Plates)**

The Ninetyeast Ridge (NER) lies in a zone between the Indian, Australian and Capricorn Plates, which result from the break-up of the vast Indo-Australian Plate along diffuse boundaries during the Neogene (e.g., Krishna et al., 2012). The current deformation regime of the NER is complex and its potential role as a plate boundary is debated. The northern part of the NER is dominated by left-lateral transpressional deformation (c.f. Sager et al., 2013). At 10 Ma, we expect this deformation to have generated a maximum differential motion between sites located on both sides of the NER of ca. 1° latitude, considering the difference of northward motion between both sides of the NER (Pubellier et al., 2003). Northward movement of the northern Ninetyeast Ridge where Site U1443 is located has paralleled that of the Indian Plate (including the Indian subcontinent) over the late Neogene (Fig. R1), thus the paleo-position of Site U1443 relative to the southern tip of peninsular India has remained relatively constant. This implies that important monsoon surface ocean currents (such as the Southwest Monsoon Current on Fig. 1c) likely had a similar influence in late Miocene waters overlying Site U1443 as they do today.

To illustrate the paleo-position and migration of the site more clearly, we now include these 9 and 5 Ma paleo-positions on all of the Figure 1 maps (yellow stars – although paleogeography in these maps is modern). This shows that, assuming modern current positions and oceanography for the Late Miocene, the seasonal contrast in mixed layer depth related to monsoon wind-driven mixing is similar at the paleo and modern positions for Site U1443.

In summary, we do not think that the late Miocene paleo-position of our site affects our interpretations related to monsoon dynamics and paleoproductivity, as the site remained north of the Equator and under the same influence of the Indian monsoon wind system (even assuming no southward shift of modern oceanographic currents, revised Fig. 1). We agree that a site migration on a similar scale in the central or eastern equatorial Pacific Ocean could have much more important consequences (e.g. migration out of the equatorial high-productivity band), but this is not the case for Indian Ocean Site U1443.We have added a sentence on paleo-position and northward migration in the "Site and Sampling" methods subsection.

I was not clear why a section on primary productivity, winds, and sediment traps were included with the drill site information. I didn't see where this was used later in the paper. If this is actually used it should be a separate subsection with a topic sentence to explain why they are making these observations. The drillsite was significantly further south when the 9-5 Ma sediments were laid down, so observations at the modern position may not be relevant.

We included a section on modern oceanography, winds, and productivity as rationale for the interpretation of our Site U1443 sedimentary paleoproductivity data as representative of summer monsoon wind strength. We think that the inclusion of this data (and description of the methods/datasets we used) is important for the paper, so that the reader understands the modern link between monsoon dynamics and export productivity in the region. We have moved this paragraph out of the "Site and Sampling" subsection of the Methods and into a separate subsection with a topic sentence, as suggested.

Although the drill site was located ~2° further south during the late Miocene study interval (see detailed response above), this does not make a large difference to monsoon-driven seasonal oceanographic changes (see revised Figure 1). We don't think it would be necessarily more relevant to extract modern data at the late Miocene paleo-locations,

because the position of Site U1443 relative to peninsular India was similar. In addition, because we compare modern oceanographic data to sediment trap data (also from 5°N), we prefer to use a box around the modern site position.

The description of the age model, XRF scans and stable isotope methods are clear. It doesn't appear that Si was independently calibrated. Is this true? How much did a ratio of Si/Ti in raw XRF counts vary down the interval, as evidence that biogenic Si deposition was negligible?

It is correct that Si was not independently calibrated. This was because the acid digestion protocol used included hydrofluoric acid resulting in the formation of $SiF_4$, which is a volatile compound. Thus, Si concentrations determined by ICP-MS are not considered to be accurate enough, so were not used for calibration.

[Figure]

The amplitude of Si/Ti XRF intensity ratio is generally small as shown by the above figure. Si/Ti remains within a relatively narrow band centred around 7 below 95 m CCSF and around 6 above 95 m (~6.8 Ma), this switch appears driven by an increase in Ti. We also note that no siliceous microfossils were observed by shipboard biostratigraphers or sedimentologists over this interval. Considering the primary sediment component is carbonates (between ~60 and 90%, generally higher than 70%, Fig. 7b) the small Si/Ti changes most likely reflect changes in the detrital fraction.

The comparison of stable isotopes to other Miocene data is clear and shows the relatively low variability of stable oxygen isotopes in this interval. One of the interesting graphs is the comparison of the stable carbon isotopes. There is a clear offset between records from different basins, but a common shape to the curve signifying a strong global signal. It is likely that stable carbon isotopes may provide a decent chronostratigraphy.

We thank the reviewer for this comment, we agree that despite inter-basin offsets the $\delta^{13}C$ trends appear synchronous. We hope our new Indian ocean records will stimulate new research on the late Miocene $\delta^{13}C$ shift.

I was puzzled why section 4.2 on XRF calibration is in the results. It clearly belongs in methods. They calibrate with a relatively small set of samples, but it seems adequate. Also, I don't understand why they didn't use the shipboard carbonates data to help calibrate the Ca record, since they were having trouble with the sediment digestion data. The Ca in clays doesn't vary a lot, so most variation in Ca is because of CaCO3. If they want to see a way to calculate CaCO3 from bulk sediment chemistry, check out Dymond et al (1976; DSDP Leg 34 Initial Reports, 575-588). The spikes in Rb and K are at the same depth in both records, so probably do represent felsic ash layers. This also shows from the raw Si data.

We have moved the description of linear correlation coefficients and % CaCO$_3$ calculation to Section 3.5 of the Methods. Section 4.2 now describes only the long-term trends in calibrated and uncalibrated XRF data. The shipboard carbonate data was performed on Hole A and a small section of Hole C (shown below). Only four samples measured for CaCO$_3$% are in core sections that were scanned for XRF (following the splice between 72.75- 113.56m CCSF), so we could not use shipboard data for calibration. We think that our chosen method for CaCO$_3$ calculation is robust, given the excellent agreement with %CaCO$_3$ calculated in the subsequent interval of the core by Lübbers et al., (2019), based on calibration of XRF-derived counts of (Ca/$\sum$(Ca, Al, Si, K, Ti, Mn, Fe, S)) to discrete CaCO$_3$ measurements (Fig. 7b).

[Figure]

Figure F16. Calcium carbonate and total organic carbon contents in sediments, Holes U1443A and U1443C.

Incidentally, how much of the total Ba was represented by the excess Ba? The productivity interpretations are more robust If the excess Ba is a large proportion of the Ba signal. I trust the spectral analysis and am heartened that the Ba-xs has a cleaner orbital signal than Ba/Fe. They would have the same signal only if Fe was constantly deposited. Otherwise there is a composite signal of both elements.

Thanks for raising this important question. $[Ba]_{xs}$ represents on average 83% of total $[Ba]$. We have added this information to the results.

Specific comments:
Line 116. Position of Site U1443 has been rounded off too much. It is OK to round to the nearest minute, not nearest degree. Actual position is 90°22'E, 5°23'N. This is important to track how the site position changed by plate tectonic motion over their time frame.

We have corrected this.

Line 215: What is CEREGE? Only the acronym is given in the address as well.

CEREGE is the name of our laboratory, Centre Européen de Recherche et d'Enseignement des Géosciences de l'Environnement. Because it is in French, we generally use only the acronym in affiliation listings. We have added the full name to the paper text.

Line 320-325: One could better judge the relative amounts of detrital Ba and bio-Ba if there were more information on percentage of clays in the interval. It would appear from descriptions that there is very little biogenic opal in the interval. If that is true, the clay content is represented by the noncarbonate fraction (100-CaCO3%). How did that vary over the interval?

Biogenic Ba constitutes on average 83% of total Ba. We confirm that we think there is very little biogenic opal in the interval, and that clay content is represented by the noncarbonate fraction (100-CaCO$_3$%). Variations in the carbonate vs non-carbonate fractions are shown in Fig. 7b/c. Over the timescale of our study, there is a small long-term increase in the clay (non-CaCO$_3$) content (also visible in the "Terrigenous MAR" record in Fig. 7, consistent with a longer-term trend of increasing mineral flux in this region of the NER from the Miocene to the Pleistocene). This is discussed in Section 5.1.

Line 365: The authors should state at the beginning that they believe their newer age model is better, for the reasons they list. When I first read this, it wasn't clear what they were claiming. Incidentally, a spectral test is not very sensitive to minor age errors. I place more credence on comparison with other tuned isotope records.

We agree that the good correspondence of our isotope stratigraphy with other, independently tuned, records provides important validation of our age model. We do think our age model for the 8.7-8.1 Ma interval is more robust than that of Lübbers et al., 2019, simply because a longer continuous isotope record was available at the time of age model construction. For example, the youngest tie-point in the age model of Lübbers et al (2019) is at 8.5 Ma and ages for the interval 8.5 to 8.1 Ma are just extrapolated using the sedimentation rate between 8.7 and 8.5 Ma.

Line 445: the MAR record is driven strongly by the age picks and only secondarily by sediment composition. Square wave profiles like seen for bulk sediment can be caused either by errors in the ages of the intervals, or by major changes in sedimentation higher than the resolution of the age model. Which do they think is the cause?

We agree that the MAR records are primarily driven by the age-depth tie points that we impose and sedimentation changes were likely less abrupt in reality, however we are confident that our age control points are robust (Fig. 4, 5,

and S1). We focus our interpretation on long-term MAR trends rather that step changes in our record. We have added the following sentence to the results: "The stepwise nature of MAR records results from age model-imposed stepped changes in sedimentation rate."

Line 500: I am having difficulty with this attempt to reconcile an increase in carbonate accumulation rate first with a decrease in dissolution but then also with an increase in productivity. The argument about scavenging is completely ad hoc. Usually there is more than enough production to remove clays from surface waters, so higher production does not lead to higher clay deposition. Furthermore, the CaCO3% and Ca/Terr records are consistent with an increase driven primarily by reduced dissolution. If there is higher clay deposition post 8.5 Ma, how can one disprove the alternative hypothesis, that of higher aeolian dust flux that may have triggered some higher production through iron fertilization or indirectly because winds were stronger and carried more dust?

Based on our records, non-CaCO$_3$ MARs (i.e., bulk MAR – CaCO$_3$ MAR, ~ clays) show a step increase of ~50% at this time, coincident with the increase in CaCO$_3$ MAR. The reviewer is right that fine-grained mineral dust, most likely from the deserts to the west bordering the Arabian Sea, could have contributed to the U1443 clay fraction. However, we consider it unlikely that wind-blown dust was a major constituent of clay at Site U1443. A recent study on detrital clay geochemistry in Site U1443 sediments (Bretschneider et al., 2021) discusses clay provenance in detail, and concludes that detrital material in Site U1443 late Miocene sediments was primarily supplied by the large river systems. One observation in support of this is the much higher clay contents at Site U1443 compared to Indian Ocean sites further from riverine detrital sources. To the best of our knowledge, there are no records documenting aeolian dust flux to the equatorial Indian Ocean spanning the Miocene. Because we cannot rule out the hypothesis that dust played a part in our record, we now raise this possibility in our discussion, and present our scavenging hypothesis as speculative. An increase in wind intensity from 8.6 Ma that could have increased aeolian dust delivery thus increasing total phytoplankton productivity and organic carbon export is not supported by our Ba records, which show no change in total export productivity over the increase in CaCO$_3$ and clay MAR.

Line 658, 659: The observation of a higher productivity regime around 11 Ma has also been observed in the eastern Pacific, in what Lyle and Baldauf (2015) referred to as the "early carbonate crash" The period between 10 and 11 Ma has the highest biogenic silica deposition of the entire record at Site U1338. This deposition interval is distinct from the late Miocene Biogenic Bloom. It is a low CaCO3 interval caused by higher opal deposition.
Thanks, we have added reference to this study.

**References cited**

Bretschneider, L., E. C. Hathorne, C.T. Bolton, D. Gebregiorgis, L. Giosan, E. Gray, H. Huang, A. Holbourn, W. Kuhnt, and M. Frank (2021). Enhanced late Miocene chemical weathering and altered precipitation patterns in the watersheds of the Bay of Bengal recorded by detrital clay radiogenic isotopes. *Paleoceanography and Paleoclimatology*: e2021PA004252.

Pubellier, M., Ego, F., Chamot-Rooke, N., & Rangin, C. (2003). The building of pericratonic mountain ranges: structural and kinematic constraints applied to GIS-based reconstructions of SE Asia. *Bulletin de la Société géologique de France*, *174*(6), 561-584.

Krishna, K. S., Abraham, H., Sager, W. W., Pringle, M. S., Frey, F., Gopala Rao, D., & Levchenko, O. V. (2012). Tectonics of the Ninetyeast Ridge derived from spreading records in adjacent oceanic basins and age constraints of the ridge. *Journal of Geophysical Research: Solid Earth*, *117*(B4).

Sager WW, Bull JM, Krishna KS. Active faulting on the Ninetyeast ridge and its relation to deformation of the Indo-Australian plate. Journal of Geophysical Research: Solid Earth. 2013 Aug;118(8):4648-68.